



**Future changes in the stratosphere-to-troposphere ozone mass flux and the contribution**
**from climate change and ozone recovery**
Stefanie Meul[1], Ulrike Langematz[1], Philipp Kröger[1], Sophie Oberländer-Hayn[1], and Patrick
Jöckel[2]
[1]Freie Universität Berlin, Berlin, Germany
[2]Deutsches Zentrum für Luft- und Raumfahrt (DLR) e.V., Institut für Physik der Atmosphäre,
Oberpfaffenhofen, Germany
**Abstract**
Model simulations consistently project an increase in the stratosphere-troposphere exchange
(STE) of ozone in the future. Both, a strengthened circulation and ozone recovery in the
stratosphere contribute to the increased mass flux. In our study, we investigate with a state-of-
the-art chemistry-climate model the drivers of future STE change as well as the change in the
distribution of stratospheric ozone in the troposphere. Our focus is on the investigation of the
changes on the monthly scale. The global mean influx of stratospheric ozone into the
troposphere is projected to increase between the years 2000 and 2100 by 53% under the RCP8.5
greenhouse gas scenario. We find the largest increase of STE in the NH in June due to
increasing greenhouse gas (GHG) concentrations. In the southern hemisphere (SH) the GHG
effect is dominating in the winter months, while decreasing levels of ozone depleting substances
(ODS) and increasing GHG concentrations contribute nearly equally to the increase in SH
summer. A large ODS-related ozone increase in the SH stratosphere leads to a change in the
seasonal breathing term which results in a future decrease of the ozone mass flux into the
troposphere in the SH in September and October. We find that the GHG effect on the STE



change is due to circulation and stratospheric ozone changes, whereas the ODS effect is
dominated by the increased ozone abundance in the stratosphere. The resulting distributions of
stratospheric ozone in the troposphere for the GHG and ODS changes differ because of the
different regions of ozone input (GHG: midlatitudes; ODS: high latitudes) and the larger
increase of tropospheric ozone loss rates due to GHG increase. Thus, the model simulations
indicate that stratospheric ozone is more efficiently mixed to lower levels if only ODS levels
are changed. The increase of the stratospheric ozone column in the troposphere explains more
than 80 % of the tropospheric ozone trend in NH spring and in the SH except for the summer
months. The importance of the future stratospheric ozone contribution to tropospheric ozone
burdens therefore depends on the season.



## 1. Introduction

Ozone ($O_3$) in the troposphere has two sources: photochemical production involving ozone
precursor species such as nitrogen oxides ($NO_x$), carbon monoxide (CO) and hydrocarbons
(e.g., methane ($CH_4$)) and the transport of ozone from the stratosphere into the troposphere (i.e.
stratosphere-troposphere exchange, STE) (IPCC, 2001). Mass can be exchanged between the
stratosphere and the troposphere along isentropic surfaces which intersect the tropopause in the
lowermost stratosphere (LMS) (Holton et al., 1995) where the chemical lifetime of ozone is
larger than the transport timescale. Tropopause folds in the vicinity of the polar and the
subtropical jets and cut-off lows are important structures for the effective transport of
stratospheric air masses into the troposphere because of their large displacements of the
tropopause on isentropic surfaces (Stohl et al., 2003). Mass exchange is also possible by slow
cross-isentropic transport, which is driven by diabatic cooling (Stohl et al., 2003) through the
large-scale vertical motion of air in the stratospheric meridional residual circulation, the
Brewer-Dobson circulation (BDC).
Earlier studies have shown that in a changing climate the mass transport from the stratosphere
will increase due to a strengthened BDC (e.g., Scaife and Butchart, 2001; Butchart et al.,
2010; Oberländer et al., 2013). Akritidis et al. (2016) found coinciding increases in the
frequency of tropopause folds in summer over the Eastern Mediterranean and in stratospheric
ozone in the lower troposphere between 1979 and 2013.  In addition to changes in the ozone
transport from the stratosphere into the troposphere, ozone concentrations in the stratosphere
are expected to change. Due to declining halogen levels in the stratosphere following the
regulation of ozone depleting substances (ODS) by the Montreal Protocol and its
amendments, stratospheric ozone is projected to recover during the 21$^{st}$ century (e.g., WMO,
2014). In addition, radiative cooling of the stratosphere associated with the rising



concentrations of well-mixed greenhouse gases (GHG) (i.e. carbon dioxide ($CO_2$), nitrous
oxide ($N_2O$) and $CH_4$) will lead to reduced ozone loss rates and an ozone increase in the
stratosphere (e.g., Jonsson et al., 2004). Both, the intensified stratospheric circulation, and the
concurrent recovery of stratospheric ozone are expected to lead to an increase in the ozone
mass entering the troposphere (e.g., Stevenson et al., 2006; Shindell et al., 2006; Hegglin and
Shepherd, 2009; Young et al., 2013; Banerjee et al., 2016). Previous studies with different
models and approaches have indicated a dominant role of stratospheric circulation changes for
the increased STE (e.g., Sudo et al., 2003; Zeng and Pyle, 2003; Collins et al., 2003). Also in
observational data, the connection between stratospheric circulation changes and tropospheric
ozone variations was identified (Neu et al., 2014). However for the past, Ordóñez et al. (2007)
showed that changes in lowermost stratospheric ozone concentrations have a larger effect on
the STE change than variations in cross-tropopause air mass transport. A reduced STE due to
stratospheric ozone depletion in the past was found by Shindell et al. (2006) to offset more
than half of the tropospheric ozone increase since preindustrial times. The influence of
stratospheric ozone recovery on STE in the future was reported by Zeng et al. (2010) who
showed that in the Southern Hemisphere (SH) during winter stratospheric ozone increase and
climate change have a nearly equal contribution to the increase in surface ozone under the
A1B scenario. More recently, the drivers of future STE changes have been analysed by
Banerjee et al. (2016) in idealized model simulations. They find that ODS and climate change
under the RCP8.5 scenario contribute about equally to the annual global STE increase
between 2000 and 2100.
Rising GHG concentrations, however, do not only affect the stratospheric circulation and
chemistry. In the troposphere, GHG-induced warming increases the water vapour content and
thus tropospheric ozone destruction (e.g., Johnson et al., 1999). This results in a decrease of



chemical ozone lifetimes (e.g., Zeng et al., 2010; Banerjee et al., 2016) which means that the
distribution and the burden of stratospheric ozone entering the troposphere are also altered.
In addition to a changing amount of stratospheric ozone in the troposphere, changing future
emissions of ozone precursor species will affect the local ozone production in the troposphere.
Large differences exist in the temporal evolution of the emissions between the Representative
Concentration Pathways (RCP) for the radiative forcing of 6.0 W/m$^2$ and 8.5 W/m$^2$
(Meinshausen et al., 2011), especially for $CH_4$. This will result in a larger ozone production
under the RCP8.5 scenario at the end of the 21$^{st}$ century compared to the RCP6.0 scenario. As
a consequence, the importance of stratospheric ozone in the troposphere in the future will
depend on the net tropospheric chemical ozone production.
Most studies addressing the question of future STE changes and their role for tropospheric
ozone trends focus on the annual and global integrated fluxes. Hegglin and Shepherd (2009)
showed also the annual cycle of the ozone mass flux derived from a boxmodel approach
introduced by Appenzeller et al. (1996). In their model simulation, the maximum ozone flux
occurs in spring in the SH and Northern Hemisphere (NH) for the 1960 to 1970 mean. In the
future (2090-2100), the peak is shifted towards late spring/early summer in the NH and towards
winter in the SH. As Roelofs and Lelieveld (1997) reported, the seasonal timing of the input of
stratospheric ozone into the troposphere is relevant for potential mixing of stratospheric ozone
towards the surface, since in summer the ozone loss rate is larger than in winter. This means
that the future distribution of stratospheric ozone in the troposphere depends not only on the
overall increase on ozone mass flux, but also on the seasonality of the input.
The aim of our study is therefore, to quantify the future changes of STE in idealized simulations
with a chemistry-climate model (CCM) under the most extreme RCP8.5 scenario for the annual
and monthly means. We identify the changes in the seasonal cycle of STE due to the projected
increase in GHGs and decline in ODS, i.e. the associated stratospheric ozone recovery.





Furthermore, we analyse the resulting changes in the distribution of stratospheric ozone in the
troposphere, using comprehensive stratospheric and tropospheric chemistry and therefore
considering the full range of changes in chemical loss and production caused by GHG or ODS
changes. The additional analysis with a transient run under the RCP6.0 scenario allows us on
the one hand to study the past changes between 1960 and 1999 and on the other hand to compare
two different scenarios and their effect on stratospheric ozone trends in the troposphere.
In this study we want to address the following research questions:
(1) How will the stratosphere-to-troposphere ozone mass flux change in the future?
(2) What are the major drivers of the future changes in the stratosphere-to-troposphere ozone
mass flux?
(3) Will the seasonality of the STE change in future?
(4) How will the GHG emission scenarios affect the ozone mass flux into the troposphere?
(5) How is the ratio of stratospheric ozone in the troposphere changed in the future?
The study is structured as followed: First the model and the experimental set-up used for the
simulations are described as well as the methodology for calculating the ozone mass flux from
the stratosphere to the troposphere (Section 2). In Section 3 we show the climatological mean
state of the year 2000 simulation for a basic evaluation. Results of mass flux changes and
changes in the distribution of stratospheric ozone in the troposphere are shown in Section 4
followed by the attribution analysis in Section 5. The results are summarized in Section 6.

## 128     2. Model experiments and methods

### 129     2.1 Model experiments

In this study we applied the EMAC (ECHAM/MESSy Atmospheric Chemistry) CCM version
described by Jöckel et al. (2016) in the T42L47MA configuration, i.e. with 47 model layers and



a horizontal resolution of 2.8°x2.8°. EMAC is a numerical chemistry and climate simulation
system that includes submodels describing tropospheric and middle atmosphere processes and
their interaction with oceans, land and human influences (Jöckel et al., 2016). It uses the second
version of the Modular Earth Submodel System (MESSy2) to link multi-institutional computer
codes. The core atmospheric model is the 5th-generation European Centre Hamburg general
circulation model (ECHAM5) (Roeckner et al., 2006). The atmospheric chemistry is calculated
using the submodule MECCA (Module Efficiently Calculating the Chemistry of the
Atmosphere; revised version by Sander et al., 2011a). The gas-phase rate coefficients follow
the latest recommendations of JPL (Sander et al., 2011b). For heterogeneous reactions in the
stratosphere the rate coefficients are calculated with the submodel MSBM (Multi-phase
Stratospheric Box Model) which also returns the parameters (e.g., number densities, surface
areas) of the sulfuric acid aerosols and the polar stratospheric cloud (PSC) particles.
To quantify the impact of increasing GHG concentrations and of declining stratospheric
halogen levels on the net ozone mass flux from the stratosphere into the troposphere, we
performed four experiments in timeslice mode, i.e. with non-varying boundary conditions from
year to year, but including a seasonal cycle. In addition to reference simulations for the years
2000 and 2100, one sensitivity simulation for GHG increase only and one for ODS decrease
only have been set up. Each timeslice simulation has been integrated over 40 years following 5
years of spin-up time. Future surface concentrations of the well-mixed GHGs ($CO_2$, $N_2O$, $CH_4$)
are prescribed according to the extreme RCP8.5 scenario (Meinshausen et al., 2011) in order to
reveal the upper boundary of the anticipated future changes. The estimated decline of ODS as
a consequence of the successful regulation of halogen containing species in the Montreal
Protocol and its amendments is given by the boundary conditions following the A1 scenario in
WMO (2011). Note that due to an unintended neglect of minor chlorine source gases CFC-113,
CFC-144, CFC-155 as well as HCFC-22, HCFC-141b and HCFC-142b, stratospheric chlorine



levels in the year 2000 are underestimated by about 10%. The quasi-biennial oscillation (QBO)
of tropical winds in the stratosphere is nudged to observations following Giorgetta and
Bengtsson (1999). Solar variability like the 11-year solar cycle is not included, instead solar
mean conditions of solar cycle number 22 are prescribed. The sea surface temperature (SST)
and sea ice concentration (SIC) fields are prescribed as 10-year averages around the respective
years using the output from a transient simulation with the coupled atmosphere ocean model
MPI-ESM (Max-Planck-Institute Earth System Model; Giorgetta et al., 2013; Schmidt et al.,
2013) for the RCP8.5 scenario. Using multi-year averages reduces the inter-annual variability
of the SSTs, but ensures quasi neutral conditions of the El Niño Southern Oscillation (ENSO).
An overview of the boundary conditions in the four simulations is given in Table 1.
To show the temporal evolution of the changes and to compare the effects for different GHG
scenarios we also analyse the model output from the transient simulation RC2-base-05 of the
Earth System Chemistry integrated Modelling (ESCiMo) project (Jöckel et al., 2016) which has
been integrated according to the RCP6.0 scenario from 1960 to 2100 following a 10-year spin-
up. The SST and SIC fields for the RCP6.0 scenario are prescribed from the Hadley Centre
Global Environment Model version 2 - Earth System (HadGEM2-ES) Model (Collins et al.,
2011; Martin et al., 2011). The boundary conditions for this simulation are given in Table 1.
More detailed information of this simulation can be found in Jöckel et al. (2016). It has to be
noted, that the stratospheric ozone loss in the past is underestimated in this simulation, which
affects the trends in ozone mass flux.

**2.2 Methods**

To quantify the net ozone mass flux from the stratosphere into the troposphere we apply the
boxmodel approach described by Appenzeller et al. (1996). They described the hemispheric net
mass transport in a simple model which consists of three regions (i.e. boxes), the troposphere,





the 'lowermost stratosphere' (LMS) and the 'overworld'. The LMS is the region where isentropic
surfaces intersect the tropopause. Thus mass can be exchanged between the stratosphere and
the troposphere along such isentropic surfaces. Above the LMS, often referred to as the
overworld according to Holton et al. (1995), the isentropic surfaces lie entirely in the
stratosphere and mass exchange is only possible by cross-isentropic transport, which is carried
out by the large-scale meridional circulation in the stratosphere (BDC). Due to mass continuity,
a mass flux from the overworld into the LMS ($F_{in}$) must be balanced by a mass flux out of the
LMS ($F_{out}$) and/or mass change (dM/dt) in the LMS:
$\quad F_{in} = F_{out} + dM/dt \qquad$ (1).
In our study we use the 91 hPa surface (nearest pressure level of the model output to 100 hPa)
as upper boundary for the LMS boxmodel and the model tropopause (a combination of the
thermal tropopause in the tropics and the dynamical tropopause in the extratropics) as lower
boundary, analogously to Hegglin and Shepherd (2009). Thus, M in Equation 1 is the total
ozone mass between these boundaries, and dM/dt is the monthly change of M. $F_{in}$ is calculated
for each hemisphere as the area-weighted integral of the product of the monthly zonal mean
ozone concentration and the negative of the monthly mean residual vertical velocity ($-\overline{w}^*$) at 91
hPa at each gridpoint. Since, by definition, the vertical velocity is positive for upward and
negative for downward motion, $\overline{w}^*$ is multiplied by the factor -1 in order to get positive values
for the downward ozone mass flux into the troposphere. Finally, $F_{out}$ is calculated as residual.
It has to be noted that with this methodology it is not possible to study the transport pathways.
The use of the 91 hPa surface as upper boundary instead of 100 hPa leads to slightly lower
values of the resulting ozone mass flux (< 2 % in the REF2000 simulation, estimation based on
linear interpolation between the two model pressure surfaces surrounding 100 hPa).
To distinguish ozone with stratospheric origin and ozone produced in the troposphere we use a



diagnostic tracer O3s (Roelofs and Lelieveld, 1997; Collins et al., 2003; Jöckel et al., 2016).
This tracer is reset in each model time step (by nudging with a relaxation time equaling the
model time step length) to the interactive ozone above the model tropopause. In the troposphere,
the chemical production of O3s is omitted while the sinks of O3s, i.e. chemical loss and dry
deposition, are considered in the same way as for O3. The loss of O3s (LO3s) plus the dry
deposition have been used as a qualitative measure of STE in earlier studies (e.g., Roelofs and
Lelieveld, 1997; Jöckel et al., 2006). However, in our study this different method of STE
estimation was not applied since the output of dry deposition for the O3s tracer was not
available. Also, LO3s is not only affected by changes in ozone flux but also by changes in
tropospheric ozone chemistry due to the prescribed forcings, potentially leading to differences
in the derived STE.

**3.   Equilibrium state of the year 2000 (REF2000)**
Before we estimate the future change in ozone mass flux into the troposphere, the present-day
equilibrium state is analysed to ensure that the important mixing processes and tracer
distributions are represented realistically in the EMAC simulations. An effective method to
investigate STE and mixing processes in the LMS are tracer-tracer correlations.
**3.1 Tracer-tracer correlations from scatter plots**
Fischer et al. (2000) introduced the $O_3$-CO correlations for aircraft in situ measurements to
analyse the chemical transition between the stratosphere and troposphere and the mixing
processes in the UTLS. Due to the sharp gradients with high CO (low $O_3$) values in the upper
troposphere and low CO (high $O_3$) values in the lower stratosphere, the $O_3$-CO scatter plot has
a characteristic L-shape in the UTLS (e.g., Fischer et al., 2000; Tian et al., 2010; Barré et al.,



2013). Pan et al. (2007) distinguish the stratospheric branch and the tropospheric branch with a
quasi linear relation between $O_3$ and CO and a transition region in between characterized by
non-linear behavior. Figure 1 (top) shows the $O_3$-CO scatter plot for the REF2000 simulation
in the northern mid-latitudes. Compared to the results shown by Fischer et al. (2000) for
measurements and Barré et al. (2013) for measurements and model data, we find that the L-
shape of the correlation is captured reasonably well by the EMAC timeslice simulation.
The shape of the $O_3$-$N_2O$ scatter plot (Figure 1, bottom) with a full stratosphere coverage results
from the negative vertical gradient in $N_2O$ and the $O_3$ maximum in the stratosphere: The
correlation is negative below the ozone maximum and positive above. The fan-shaped structure
is due to the horizontal gradient with higher $O_3$ and $N_2O$ values in the tropics than in the extra-
tropics. The model result is in qualitative agreement with $O_3$-$N_2O$ scatter plots for ACE
measurements shown by Hegglin and Shepherd (2007).
This comparison indicates that the dynamical and chemical processes in the transition region
between the troposphere and stratosphere are realistic in the EMAC timeslice simulations and
allows us to assess the future changes.
**3.2 Ozone mass flux**
The annual cycle of the ozone mass flux into the troposphere ($F_{out}$, calculated according Eqation
1) for the year 2000 reference simulation is shown in Figure 2a integrated globally and over the
NH and SH respectively. The ozone mass flux into the NH is larger than into the SH and has
its peak in early summer, whereas in the SH the maximum ozone mass flux is found in spring.
The annual cycle in the EMAC REF2000 simulation is comparable to the results of Hegglin
and Shepherd (2009) for the period 1960 to 1970, but with a less pronounced peak in the NH
spring in EMAC.



Integrated over all months, the ozone mass flux reaches $390 \pm 17$ Tg/year in the NH, $322 \pm 15$
Tg/year in the SH and $712 \pm 24$ Tg/year globally. The global ozone mass flux hits the upper
boundary of the ozone mass flux derived from tropospheric models for the year 2000 ($552 \pm$
168 Tg/year; Stevenson et al., 2006) and is larger than the ozone mass flux reported by Hegglin
and Shepherd (2009) ($655 \pm 5$ Tg/year), which was derived by averaging the 1995 to 2005
period of a transient simulation with a middle atmosphere resolving CCM. However, the ozone
mass flux in the REF2000 simulation lies in the range of 340 – 930 Tg/year given by Collins et
al. (2000) for a range of models and agrees well with the ozone mass flux of $770 \pm 400$ Tg/year
given in IPCC (2001). Compared to estimates derived from observations ($500 \pm 140$ Tg/year;
Olsen et al., 2002), the ozone mass flux in the EMAC timeslice simulation is slightly
overestimated.
To better understand the changes in the calculated ozone mass flux ($F_{out} = F_{in} - dM/dt$), we
analyse the climatological annual cycle of the two ozone mass flux components, $F_{in}$ at 91 hPa
(Figure 2b) and $-dM/dt$ (Figure 2c). The ozone mass flux across the 91 hPa pressure level is
controlled by the seasonality of the BDC with the maximum (ozone) mass transport in the
winter hemisphere and a hemispherically asymmetric strength. The ozone distribution in the
stratosphere (see also Figure 3a) with low columns in the tropics and high columns in the middle
and high latitudes also reflects the structure of the stratospheric meridional circulation.
The seasonal breathing of the LMS leads to a shift of the maximum ozone mass flux from winter
to spring (Hegglin and Shepherd, 2009). The amplitude of the seasonal cycle in $-dM/dt$ is
slightly larger in the NH than in the SH which dampens the amplitude of the ozone mass flux
in the NH.
The timing of the maximum ozone mass flux into the troposphere is relevant for the resulting
downward mixing since the chemical lifetime of tropospheric ozone in the mid- and high




latitudes has a pronounced seasonal cycle with short lifetimes in the summer and relatively long
lifetimes in winter and spring. This means that ozone can be mixed more efficiently with
tropospheric air masses in winter and spring (Roelofs and Lelieveld, 1997) although the ozone
influx from the stratosphere is smaller than in summer.
In the next section we analyse the abundance of stratospheric ozone in the troposphere for June
(Figure 3), when the ozone mass flux is maximal in the NH and minimal in SH (Figure 2a).

### 3.3 Stratospheric ozone in the troposphere

In the troposphere, the columns of ozone originating from the stratosphere (Figure 3b) reach a
maximum of 30 DU around 30° in both hemispheres and a minimum in the tropics (3-6 DU
over Indonesia) and the southern high latitudes. The low values in the tropics presumably result
from very short ozone lifetimes near the surface as the high insolation and high water vapour
concentrations in the intertropical convergence zone (ITCZ) form a strong sink for tropospheric
ozone (Roelofs and Lelieveld, 1997). In addition, the downward transport of stratospheric
ozone in the tropics is very small due to the upward branch of the BDC in this region. The high
O3s columns in the subtropics result from high abundances of stratospheric ozone in the upper
troposphere, especially in the NH subtropics, which is evident in Figure 3c. Here, ozone
entering the troposphere through tropopause folds is efficiently transported to lower altitudes
in the downward branch of the Hadley cell (Roelofs and Lelieveld, 1997) resulting in relatively
high O3s levels in the middle troposphere around 30°N. The O3s mixing ratios decrease with
lower altitude and reach their minimum near the surface, with the smallest values in the tropics
and the NH, since ozone loss is largest in summer (Figure 3d). However, ozone originating
from the stratosphere is also found down to the lower troposphere in the extra-tropics. This may
be caused by events when stratospheric air penetrates deep into the troposphere and affects also
lower levels (e.g., Škerlak et al., 2014).



The contribution of O3s to ozone is in the range between 20 % in the tropics and the NH lower
troposphere and up to 40 % in the middle troposphere (500 hPa) in the NH. In the SH, the
relative contribution of O3s to $O_3$ is larger (50-60 % near the surface and more than 60 % at
500 hPa at high latitudes). This is caused by low chemical loss of O3s in winter (Figure 3d) in
combination with small chemical production of tropospheric ozone in this season. In SH
summer this pattern is reversed (not shown), however, with a slightly larger contribution of O3s
near the SH surface (20-30 %) compared to NH summer. This is possibly related to the lower
local photochemical ozone production in the SH due to reduced emissions and abundances of
tropospheric ozone precursor species.
In summary, we have found realistic tracer distributions in the tropopause region of the EMAC
reference simulation for the year 2000. The ozone mass flux appears to be overestimated
compared to observations and other model studies, while lying within the range estimated in
IPCC (2001). Given the large uncertainties for estimates from observational data and the range
of different model types, the ozone mass flux in EMAC can be regarded as reasonable. The
results indicate that the important processes determining the STE are sufficiently well
reproduced by EMAC, which allows us to study in the next section the past and future changes
of the ozone mass flux as well as the contributions from GHG and ODS changes.

**4. Past and future changes in ozone mass flux into the troposphere**
Changes in the input of stratospheric ozone into the troposphere can be caused by changes in
the dynamical processes and/or by the amount of ozone that is available for transport in the
stratosphere. Thus, not only GHG concentrations may have an impact on the stratosphere-to-
troposphere transport but also the development of ODS. The temporal evolution between 1960
and 2099 of the integrated ozone mass flux for the RCP6.0 simulation, which includes both, the



observed, and projected ODS and GHG changes, is shown in Figure 4. In the past (1960-1999),
the integrated ozone mass flux exhibits a negative trend in both hemispheres with a larger
change of -1.4 %/decade in the SH which is in qualitative agreement but smaller than the trend
(-2.3 %/decade) found by Hegglin and Shepherd (2009) between 1965 and 2000. Zeng et al.
(2010) showed that this negative trend is associated with the ODS-induced ozone loss in the
stratosphere which is most prominent in the southern polar region in spring. Between 2000 and
2099 the ozone mass flux is projected to increase globally by 4.2 %/decade. Again, the change
in the SH (4.9 %/decade) is slightly larger than in the NH (3.7 %/decade). This increase may
be the consequence of different forcings: (1) the regulations of ODS emissions lead to a decline
of chlorine in the stratosphere and increasing stratospheric ozone levels; (2) the increasing GHG
concentrations alter the temperature structure of the atmosphere and intensify the large-scale
mass transport in the stratosphere, and (3) the radiative cooling of the stratosphere due to
increasing GHG concentrations slows chemical loss reactions, which increases the ozone
amount in the stratosphere. To understand the impact of ODS and GHG changes on the ozone
mass flux in more detail, we further analyse the sensitivity simulations following the RCP8.5
scenario. For comparison the reference timeslice simulations for the years 2000 and 2100 are
included in Figure 4. The 1995-2004 average in the RCP6.0 simulation gives an ozone mass
flux of $688 \pm 24$ Tg/year which is slightly lower than in the timeslice simulation for 2000, but
within the range of two standard deviations of the REF2000 simulation. This difference might
be due to the reduced variability in the timeslice simulation compared to the transient one (see
Section 2), and/or due to the different SST/SIC fields used in the simulations. However, the
results of the model simulations are in relatively good agreement.
In the future, the ozone mass flux is clearly larger in the timeslice simulations than in the
transient simulation due to the more extreme GHG emission scenario (RCP8.5 compared to
RCP6.0). The integrated ozone mass flux reaches $598 \pm 29$ Tg/year in the NH, $490 \pm 23$ Tg/year



in the SH and 1088 ± 43 Tg/year for the global sum. This corresponds to a relative increase of
5.3, 5.2 and 5.3 %/decade, respectively (see also Table 2). Thus, in contrast to the transient
RCP6.0 simulation, the future ozone mass flux change in the RCP8.5 timeslice simulations is
similar in the NH and in the SH.

**5.  Attribution of future changes in ozone mass flux to climate change**
Figure 5 shows the monthly changes of the ozone mass flux for the sensitivity simulations, i.e.
for the total change between 2000 and 2100 due to all forcings and the contributions from GHG
and ODS changes. It has to be noted that the changes due to GHGs and ODS do not necessarily
sum up to the total change because of non-linear interactions and the missing change in
tropospheric ozone precursor species in the ODS-only and GHG-only simulations (see Table

359    1).

The change in ozone mass flux between 2000 and 2100 due to all forcings (top row) is positive
throughout the year with maximal increases in the summer months of the respective
hemispheres by up to 32 Tg/month (75 %) in the NH and 19 Tg/month (68 %) in the SH. The
GHG (middle row) and ODS (bottom row) induced changes clearly indicate the dominant role
of rising GHG concentrations for the future ozone mass flux change in the NH, explaining 80
to 95 % of the total change. The GHG-related ozone mass flux increase in the NH is maximal
in June and July, slightly shifting the peak in the annual cycle to summer which is consistent
with the findings by Hegglin and Shepherd (2009) for the total change between the 1960-1970
and 2090-2100 means. The ODS decrease, however, leads only to small positive and (not
significant) negative changes in the NH.



In the SH, the GHG-induced increase dominates the ozone mass flux change in winter and
spring, but in summer the ODS-related increase of the ozone mass flux contributes up to 50 %
to the total change in the SH. A significant reduction of ozone mass flux is found from August
to October in the SH due to the ODS change. This causes a shift of the SH maximum ozone
flux from October to January and is in contrast to the results by Hegglin and Shepherd (2009)
who found the maximum SH ozone mass flux in the future (2090-2100) to occur in August.
Overall we find that the GHG-induced changes will determine the positive trend of the ozone
mass flux in the NH, while in the SH both ODS and GHG changes affect the trend and the
seasonality of the future ozone mass flux into the troposphere.
To identify the processes behind the ODS- and GHG-induced changes, we analyse the changes
of the two ozone mass flux components, i.e. $F_{in}$ and the seasonal breathing term. We find that
$F_{in}$ will increase in the future throughout the year in both hemispheres and for both external
forcings. Figure 6 shows in the top row the latitudinal distribution of the product of ozone
concentration and $-\overline{w}^*$ at 91hPa, which equals $F_{in}$, when integrated over all latitudes. The two
components of $F_{in}$, $-\overline{w}^*$ and the ozone concentration, are shown separately in the middle and
bottom rows of Figure 6, respectively. The increase of $F_{in}$ (or $O_3$ x $-\overline{w}^*$) due to the GHG effect
(Fig. 6d) is caused by an increase in the downwelling (i.e. $-\overline{w}^*$, positive for downwelling, Fig.
6e) of the BDC in the winter season with climate change (e.g., Sudo et al., 2003; Butchart et
al., 2010; Oberländer et al., 2013) in combination with an ozone increase resulting from
stratospheric cooling and enhanced meridional transport (Figure 6f). In contrast, with ODS
decrease no significant changes in the downwelling occur (Figure 6h). The small increase in $F_{in}$
(Figure 6g) is therefore attributed to stratospheric ozone recovery from ODS, in particular in
Antarctic spring (Figure 6i). Figure 6 also indicates that the maximum change in ozone mass
flux into the troposphere occurs at midlatitudes for the GHG increase and at high latitudes for



the ODS reduction. This may have an influence on the mixing and distribution of stratospheric
ozone in the troposphere (see below).
Thus, given the positive changes in $F_{in}$, the significant negative change in the ozone mass flux
identified in September and October for the ODS decrease, must be attributed to changes in –
$dM/dt$ (i.e. the monthly change in the ozone mass contained in the LMS, also referred to as
seasonal breathing). While the total mass in the LMS is decreasing with rising GHG
concentrations in the sensitivity simulations (possibly due to the tropopause lifting effect of
rising GHGs), it slightly increases with ODS change only (not shown). The mass of ozone (M),
however, is increasing globally due to both, GHGs and ODS. Thus, for the GHG effect, the
future increase of ozone in the LMS outweighs the reduction in total LMS mass. If this future
increase of M in the LMS is monthly varying, a future change (positive or negative) in $-dM/dt$
will result. Exactly this is happening in SH spring, when the ozone mass increase is steadily
amplified between August and November due to the decline of ODS. This results in the shift of
the seasonality of the ozone mass flux and therefore to negative changes in SH spring.
As mentioned above, the timing of the strongest input of stratospheric ozone into the
troposphere is relevant in that the efficiency of mixing down to lower altitudes or to the surface
depends on the chemical lifetime of ozone which varies with season. A shift of the spring
maximum in the SH to summer (January) for instance may result in different mean abundances
of O3s in the troposphere. Furthermore, the chemical loss of ozone will increase in a warmer
troposphere, affecting the lifetime of ozone and thus the distribution of stratospheric ozone in
the troposphere.
The future changes in the distribution of O3s mixing ratios are shown in Figure 7 for June. O3s
is projected to increase throughout the extra-tropical troposphere. The largest changes will
occur in the subtropics in the upper and middle troposphere, the regions where cross-tropopause





transport along isentropic surfaces is possible and ozone is efficiently transported into the
troposphere through tropopause folds. This pattern is caused by the rising GHG concentrations
(Figure 7b). Near the surface however, the O3s mixing ratios will decrease with GHG change
at summer NH mid-latitudes, either induced by an increased chemical O3s loss or dry
deposition. The total positive change near the surface results from the O3s increase due to ODS
change (Figure 7c). In the SH, the abundance of stratospheric ozone increases throughout the
troposphere down to the surface. More O3s seems to be transported further down than in the
NH, which may be related to the longer chemical lifetime of ozone in winter. This is also
obvious from the ODS-induced changes, albeit with very small signals.
The annual mean column-integrated values of O3s and ozone in the troposphere and their
respective changes are listed in Table 3. The O3s column increases globally by 42 % between
the years 2000 and 2100, with a larger change occurring in the SH than in the NH. Consistent
with the results above, the main contribution is from the GHG changes, whereas the ODS
changes have the largest effect on the SH (+ 6 %). These changes may result from the
combination of an increased/monthly shifted ozone mass flux into the troposphere, increased
chemical loss of O3s and changes in dry deposition of O3s. The increase in the total burden of
tropospheric ozone between 2000 and 2100 (derived from Table 3) indicates that the main
contribution to the change is from O3s (19 %), whereas ozone produced in the troposphere
(calculated as residual) causes an increase of 12 % summing up to a total increase of 31 % of
tropospheric ozone. The larger increase in the chemical loss of O3s compared to the increase in
the ozone mass flux indicates changing chemical conditions in the troposphere due to climate
change. This means that the larger amount of stratospheric ozone entering the troposphere does
not accumulate to the equivalent larger abundance of O3s in the troposphere.



Next we investigate, to what extent the future change in O3s (Figure 8, middle row) contributes
to the ozone change (Figure 8, top row) in the troposphere. The relative contribution is shown
in Figure 8 (bottom row) as annual cycle of the tropospheric columns for (c) the change between
2000 and 2100 due to all forcings, (f) the respective change due to GHGs, and (i) the respective
change due to ODS. We find that at SH middle and high latitudes more than 80 % of the increase
in tropospheric ozone column is caused by ozone originating from the stratosphere from April
through October. A similar strong contribution to the overall change of more than 80 % occurs
in the NH extratropics, however confined to the spring season (March, April and May). For the
rest of the year, ozone originating from the stratosphere causes more than 50 % of the total
change in both hemispheres. In contrast, in the tropics only 20 to 50 % of the ozone change are
attributable to changes in ozone from the stratosphere throughout the year.
In addition, our simulations illustrate that the future enhancement of stratospheric ozone import
into the troposphere and the resulting tropospheric ozone change will be dominated by the GHG
effect. If only the concentrations of ODS would decline between the years 2000 and 2100, a
minor increase in tropospheric ozone burden in the (mainly SH) extratropics would form
(Figure 8g), which is almost completely attributable to increased stratospheric ozone entering
into the troposphere (Figures 8h, 8i). However, only in the simulation with increased GHG
concentrations, the patterns and the amount of tropospheric ozone increase (Figure 8d) and the
contribution of stratospheric ozone to this increase (Figure 8e), as shown in the simulation with
all forcings (Figures 8a, 8b), are well reproduced. Up to 80 % of the tropospheric ozone trends
in SH winter and 70 % in NH spring can be explained by increased abundances of stratospheric
ozone due to the GHG effect (Figure 8f). These numbers also indicate the strong increase of
tropospheric photochemical ozone production in the future due to the doubling of methane
emissions under the RCP8.5 scenario (e.g., Young et al., 2013; Meul et al., 2016). In NH
summer, about 50 % of the change are due to stratospheric ozone, while in the tropics and the



SH summer months, the contribution is less than 40 %. This reflects the effect of the
substantially increased ozone loss rates resulting from the more tropical/subtropical downward
transport of stratospheric ozone with enhanced GHG concentrations. The chemical ozone loss
rate in the troposphere in the ODS simulation is less influenced and nearly unchanged in the
tropics. In summary, Figure 8 shows that the input of stratospheric ozone is the dominant driver
of ozone changes in the troposphere, if only ODS levels are reduced. For the GHG increase we
find that other processes, such as tropospheric chemistry, modulate the tropospheric ozone
abundance in addition to the increased influx of stratospheric ozone.
Finally, we compare the tropospheric O3s columns derived from the timeslice simulations
under the RCP8.5 scenario with the transient simulation using the RCP6.0 scenario. Figure 9a
shows the evolution of annual mean tropospheric ozone (solid) and O3s (dashed) columns for
the NH and the SH. Tropospheric ozone increases in the RCP6.0 simulation from 1960 to the
middle of the 21$^{st}$ century and slightly declines afterwards in the NH, while it stays nearly
constant in the SH. There is very good agreement of the tropospheric ozone column between
the transient and timeslice simulations for the year 2000, when both simulations use observed
GHG concentrations. Regarding the temporal evolution of O3s, we find a positive trend in both
hemispheres and only a slight decrease in the NH at the end of the 21$^{st}$ century. In the past, an
effect of the ODS driven stratospheric ozone loss is overlaid by the GHG related increase in
both hemispheres. However, a slightly smaller rise of O3s in the SH might be an indication.
The RCP8.5 scenario (circles) leads to higher values of tropospheric ozone in 2100 which is
related to two effects: a larger import of stratospheric ozone and a larger chemical ozone
production in the troposphere due to strongly enhanced methane concentrations in the second
half of the 21$^{st}$ century in the RCP8.5 scenario (see Meinshausen et al., 2011).





The ratio between O3s and tropospheric ozone (Figure 9b) gives an indication if the role of
stratospheric ozone in the troposphere will change in the future. In the past period of the
transient simulation (i.e. RCP6.0 scenario), the relative contribution of O3s decreases from 48
% (1960s) to 44 % (1990s) in the NH and from 52 % to 48 % in the SH. This is caused by an
increase of ozone produced in the troposphere, which is stronger than the increase of O3s
(Figure 9a). In the future, however, the relative importance of ozone from the stratosphere
increases, reaching 49 % in the NH and 55 % in the SH around the year 2100. Thus, in the
RCP6.0 scenario (more than) half of the ozone in the troposphere will originate from the
stratosphere in the (SH) NH at the end of the 21st century.
The comparison with the timeslice simulation (RCP8.5 GHG scenario) shows that the
abundance of O3s in the troposphere is lower in the year 2000 than in the transient simulation.
This is probably caused by the different data sets used for the SST/SIC fields in the timeslice
and transient simulation (see Table 1) leading to different tropopause heights and therefore to
different tropospheric columns. However, the larger contribution of O3s to ozone in the SH (48
%) compared to the NH (43 %) is confirmed. In 2100, tropospheric ozone columns in the NH
(SH) will consist to 46 % (52 %) of ozone originating from the stratosphere. Thus, the increase
in the contribution of O3s in the future is slightly smaller in the RCP8.5 scenario than in the
RCP6.0 scenario, despite the larger increase in ozone mass flux shown in Figure 2. Here, the
different evolution of tropospheric ozone production in the two GHG scenarios plays a crucial
role.

**6. Summary**
In this study we have analysed the future changes in stratosphere-to-troposphere transport of
ozone in timeslice and transient simulations with the CCM EMAC to address the questions





brought up in the introduction: (1) How will the stratosphere-to-troposphere ozone mass flux
change in the future? (2) What are the major drivers of the future changes in stratosphere-to-
troposphere ozone mass flux? (3) Will the seasonality of the STE change in future? (4) How
will the GHG emission scenarios affect the ozone mass flux into the troposphere? (5) How is
the ratio of stratospheric ozone in the troposphere changed in the future?
In agreement with other studies (e.g., Sudo et al., 2003; Collins et al., 2003; Hegglin and
Shepherd, 2009; Banerjee et al., 2016), we find that the influx of stratospheric ozone into the
troposphere will increase in the future. Between 2000 and 2100 the EMAC timeslice
simulations project an increase of the annual global mean ozone mass flux by 53 % under the
RCP8.5 scenario. Increasing GHG concentrations were identified as the main driver of the
rising ozone mass flux into the troposphere by strengthening the BDC and reducing chemical
ozone loss in a colder stratosphere. The annual global ozone mass flux is increased by 46 %
due to rising GHG concentrations compared to an increase of 7 % due to the ODS decline and
the associated ozone recovery. The GHG effect leads to a larger intensification of STE in the
NH (51 %) than in the SH (40 %), whereas the ODS effect is most prominent in the SH (9 %)
compared to 4 % in the NH.
Regarding the seasonal changes of the ozone mass flux, we showed the dominant role of GHG
changes for the NH whereas in the SH, both ODS and GHG changes affect the seasonality of
the ozone mass flux increase: the GHG increase is the main driver of the increase in winter and
spring, but in summer also ODS-induced changes contribute to the ozone mass flux increase.
Furthermore, the ODS decrease and the concomitant ozone increase in the lower stratosphere
during SH spring cause a large change in the seasonal breathing term in the SH from August to
October, which results in a shift of the maximum ozone flux to late spring/early summer. The
GHG effect leads to a dampened amplitude of the seasonal cycle in the SH and an intensified



in the NH. This can have an impact on the distribution of stratospheric ozone in the troposphere:
in the SH more ozone is transported into the troposphere in winter, when the chemical lifetimes
are relatively long, whereas in the NH the largest increase is found in summer. This may explain
the larger increase of O3s columns in the SH compared to the NH despite the smaller increase
in ozone mass flux (see Table 2 and 3).
The future spatial distribution of the tropospheric O3s column in the troposphere is determined
by the change pattern due to GHG increases. Here, the largest increase of O3s mixing ratios
occurs in the subtropical upper troposphere, where stratospheric ozone is transported into the
troposphere via tropopause folds and then further down to lower levels in the large-scale sinking
of the Hadley cell (Roelofs and Lelieveld, 1997). ODS-related changes in the tropospheric O3s
column are smaller. They show no comparable signal in the subtropical region, but a more
homogeneous distribution. In the ODS simulation, the main increase of stratospheric ozone
input occurs via the downward branch of the BDC in middle and higher latitudes, where the
chemical ozone loss of tropospheric ozone is smaller than in the subtropics and hence mixing
towards the surface is more efficient.
In the SH winter months, the ozone change due to increased stratospheric ozone influx explains
up to 80 % of the overall tropospheric ozone increase under the RCP8.5 scenario by the end of
the century. In the rest of the year, the stratospheric ozone changes cover more than 50 % of
the ozone changes in the SH troposphere. In contrast, increased stratospheric ozone explains
only about 70 % of the ozone changes in NH spring indicating the strong increase of
tropospheric photochemical ozone production in the future due to the doubling of methane
emissions under the RCP8.5 scenario.
The comparison with the transient EMAC simulation under the RCP6.0 scenario shows a
smaller future increase in annual global ozone mass flux into the troposphere (4.2 %/decade)



than under the RCP8.5 scenario (5.3 %/decade). In the transient RCP6.0 simulation the positive
trend between 2000 and 2100 is larger in the SH than in the NH, which is not found in the
RCP8.5 timeslices. The stronger increase in the ozone mass flux under the RCP8.5 scenario is
connected with a larger O3s column, but the relative contribution of O3s to ozone in the
troposphere rises similarly in both scenarios. This is caused by the different evolution of the
ozone produced in the troposphere in the RCP6.0 and RCP8.5 scenario. In the past, the input of
stratospheric ozone has slightly decreased between 1960 and 1999, especially in the SH (-1.4
%/decade) due to the formation of the ozone hole. However, the O3s column in the troposphere
integrated over the NH and the SH shows a small positive trend. This may be related with the
seasonal timing of the changes, since ozone loss in the SH stratosphere has the largest effect on
the ozone mass flux in spring and early summer when the tropospheric ozone loss rates are
higher than in winter and mixing is less efficient anyway.
In summary, this study shows that GHG and ODS changes have different effects on the future
ozone mass flux, the seasonality and the resulting abundances of stratospheric ozone in the
troposphere. Moreover, it shows that both forcings are projected to cause an increased amount
of stratospheric ozone in the troposphere, which will not only contribute to the radiative forcing
and global warming but will also affect the air quality at the surface.

**Code availability**
The Modular Earth Submodel System (MESSy), including the EMAC model, is continuously
further developed and applied by a consortium of institutions. The usage of MESSy and access
to the source code is licensed to all affiliates of institutions, which are members of the MESSy
Consortium. Institutions can become a member of the MESSy Consortium by signing the



MESSy Memorandum of Understanding. More information can be found on the MESSy
Consortium website (http://www.messy-interface.org).

## Data availability

The data of the ESCiMo simulation RC2-base-05 will be made available in the Climate and
Environmental Retrieval and Archive (CERA) database at the German Climate Computing
Centre (DKRZ; http://cera-www.dkrz.de/WDCC/ui/Index.jsp). The corresponding digital
object identifiers (doi) will be published on the MESSy Consortium web page
(http://www.messy-interface.org). A subset of the RC2-base-05 simulation results has been
uploaded to the BADC database for the CCMI project. Data of the EMAC timeslice simulations
performed for this for this paper are available at the Freie Universität Berlin on the SHARP
data archive under ACPD_ozone_transport_Meul_et_al_2018.tar.

## Author contribution

SM has performed and analysed the timeslice simulations and has written the manuscript. UL
has initialized the study and has considerably contributed to the manuscript and the discussion.
PK has contributed to the analysis of the model data. SOH has performed the timeslice
simulations and has contributed to the discussion of the results. PJ led the ESCiMo project,
coordinated the preparation of the EMAC simulation setups and conducted the model
simulations (here RC2-base-05). Moreover, he contributed to the EMAC model development,
including the here applied O3s diagnostics.



**Competing interests**

The authors declare that they have no conflict of interest.

**Special issue statement**

This article is part of the special issue "The Modular Earth Submodel System (MESSy) (ACP/GMD inter-journal SI)". It is not associated with a conference.

**Acknowledgements**

This work has been funded by the Deutsche Forschungsgemeinschaft (DFG) within the DFG Research Unit FOR 1095 "Stratospheric Change and its Role for Climate Prediction" (SHARP) under the grants LA 1025/13-2 and LA 1025/14-2. The authors are grateful to the North-German Supercomputing Alliance (HLRN) for providing computer resources and support. The EMAC model simulation RC2-base-05 was performed at the German Climate Computing Centre (DKRZ) through support from the Bundesministerium für Bildung und Forschung (BMBF). DKRZ and its scientific steering committee are gratefully acknowledged for providing the HPC and data archiving resources for the projects 853 (ESCiMo – Earth System Chemistry integrated Modelling).

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





**Table 1**. EMAC CCM simulations used in this study.

| Name | Run mode | GHG | Tropos. O3 precursor | ODS | SST/SIC |
|---|---|---|---|---|---|
| RCP6.0 (referred to as RC2-base-05 in Jöckel et al., 2016) | Transient (1960-2099, after 10 years spinup) | RCP6.0 | RCP6.0 | Observations and A1 | HadGEM 1960-2099 |
| REF2000 | Timeslice (40 years, after 5 years spinup) | Observations for 2000 | Observations for 2000 | Observations for 2000 | MPI-ESM 1995-2004 |
| REF2100 | Timeslice (40 years, after 5 years spinup) | RCP8.5 for 2100 | RCP8.5 for 2100 | A1 for 2100 | MPI-ESM 2095-2104 |
| GHG2100 | Timeslice (40 years, after 5 years spinup) | RCP8.5 for 2100 | Observations for 2000 | Obs. for 2000 | MPI-ESM 2095-2104 |
| ODS2100 | Timeslice (40 years, after 5 years spinup) | Observations for 2000 | Observations for 2000 | A1 for 2100 | MPI-ESM 1995-2004 |



**Table 2**. Overview of the annual ozone mass flux into the troposphere and the corresponding
standard deviations in the EMAC timeslice simulations. Gray numbers indicate the change
relative to the REF2000 simulation.

| | O3 mass flux [Tg/yr] | | |
|---|---|---|---|
| | global mean | NH | SH |
| REF2000 | 712±26 | 390±18 | 322±16 |
| REF2100 | 1088±43 <br> +53% | 598±29 <br> +53% | 490±23 <br> +52% |
| GHG2100 | 1041± 36 <br> +46% | 590±28 <br> +51% | 451±26 <br> +40% |
| ODS2100 | 758±26 <br> +7% | 406±20 <br> +4% | 352±13 <br> +9% |







**Table 3**. Overview of the annual mean tropospheric O3 and O3s burden [Tg] with the
corresponding standard deviations in the EMAC timeslice simulations. Grey numbers indicate
the change relative to the REF2000 simulation.

|  | Tropospheric O3 column [Tg] | | | Tropospheric O3s column [Tg] | | |
|---|---|---|---|---|---|---|
|  | global mean | NH | SH | global mean | NH | SH |
| REF2000 | **401±2** | **222±2** | **179±1** | **182 ±3** | **96 ±2** | **86 ±2** |
| REF2100 | **527±3** +31% | **290±2** +31% | **237±2** +32% | **258 ±4** +42% | **134 ±2** +40% | **123 ±2** +43% |
| GHG2100 | **513±3** +28% | **282± 2** +27% | **231±2** +29% | **246 ±3** +35% | **128 ±2** +33% | **118 ±2** +37% |
| ODS2100 | **409±2** +2% | **225±2** +1% | **184±1** +3% | **189 ±2** +4% | **98 ±1** 2% | **91 ±1** +6% |
























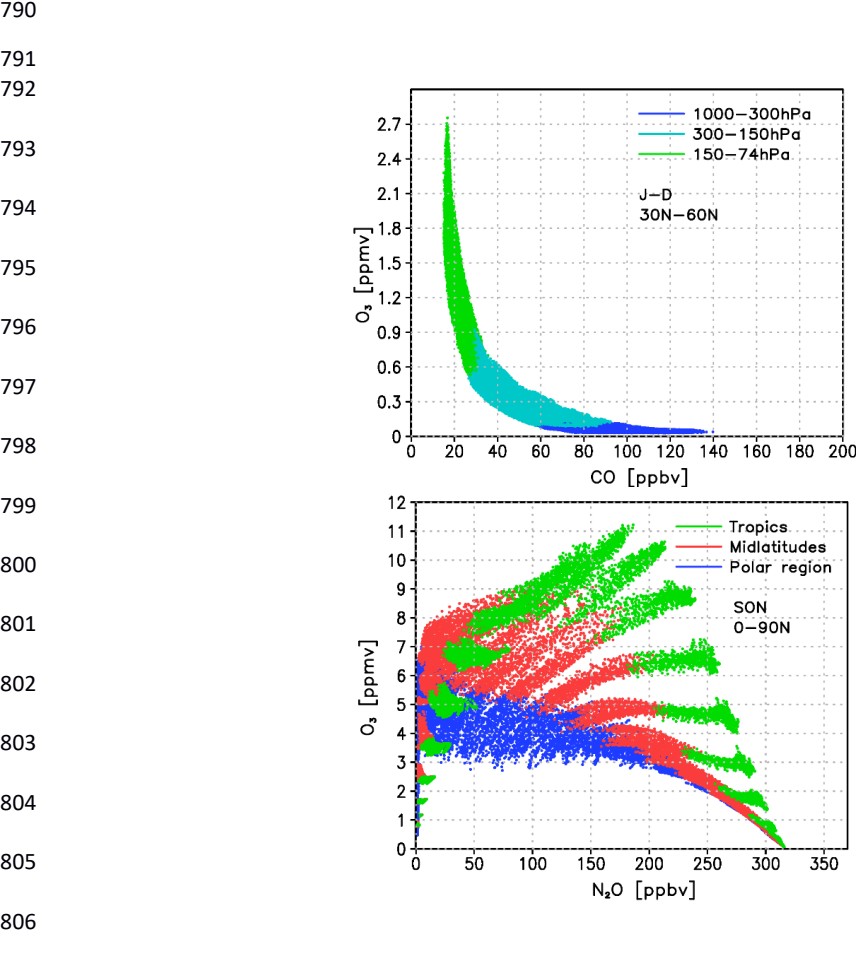

**Figure 1**: Top: $O_3$-CO scatter plot for the months January to December in the latitude band
30°-60°N for the REF2000 simulation from the troposphere to the lower stratosphere. Color
coding indicates different height regions. Bottom: $O_3$-$N_2O$ scatter plot for the months
September to November in the Northern Hemisphere for the REF2000 simulation for the
altitude region between 270 and 0.1 hPa. Color coding indicates different latitude bands.






















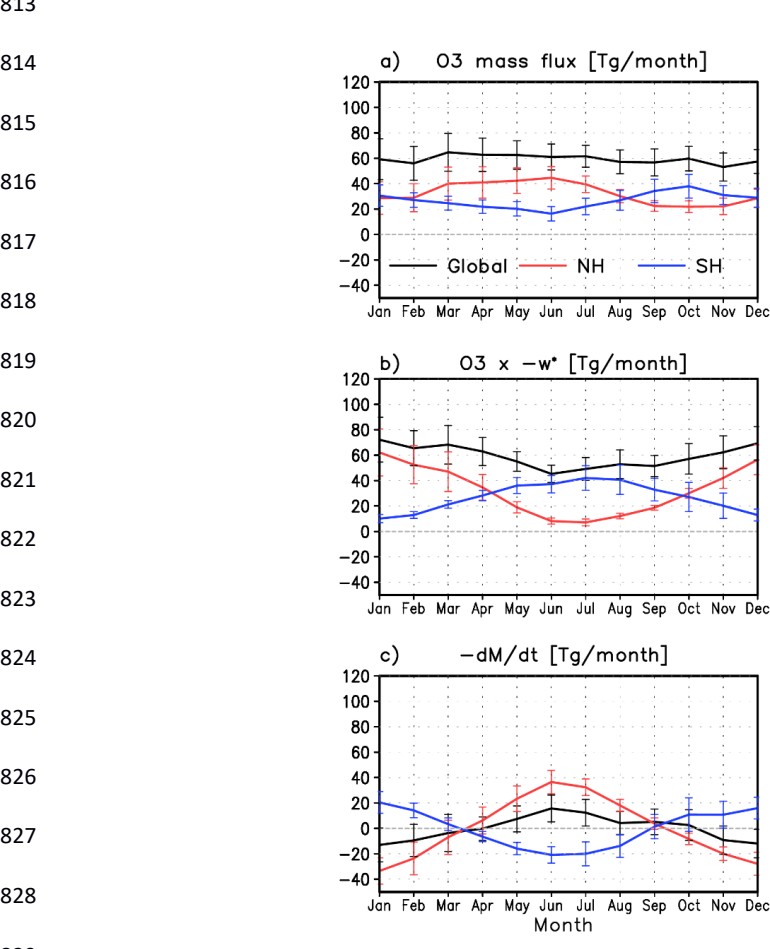

**Figure 2**: a) Annual ozone mass flux and its 95% confidence interval (i.e. $\pm 2\sigma$, with $\sigma :=$ standard deviation) [Tg/month] from the stratosphere into the troposphere ($F_{out}$) in the REF2000 simulation integrated globally (black), over the northern (red) and southern (blue) hemispheres. b) As a) but for $F_{in}$ (the product between the ozone concentration and the negative zonal mean residual vertical velocity $\bar{w}^*$ at 91 hPa). c) As a) but for the negative monthly change in ozone mass of the LMS (-dM/dt), also referred to as seasonal breathing.










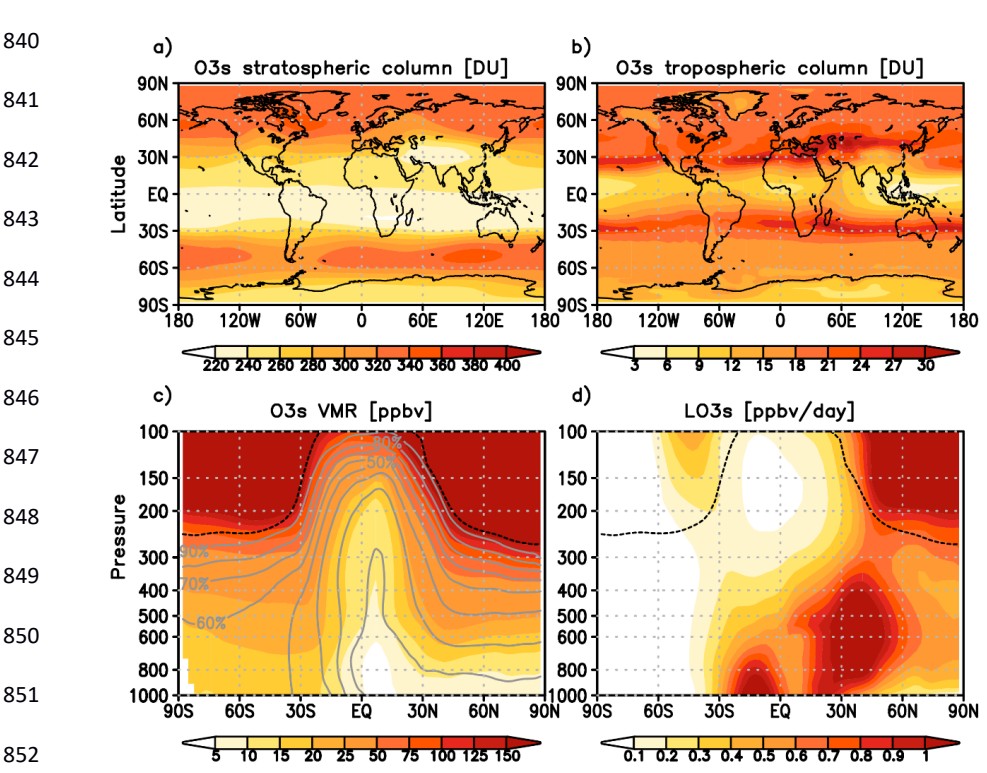

**Figure 3**: a) Geographical distribution of stratospheric partial columns of the diagnostic O3s
tracer in Dobson Units (DU) in June for the REF2000 simulation. b) as a) but for the
tropospheric columns. c) Latitude-height section of the O3s volume mixing ratios [ppbv] and
d) latitude-height section of the chemical loss rate of O3s [ppbv/day]. The black dashed line
indicates the position of the mean tropopause. Gray contour lines in c) show the relative
contribution of O3s to the ozone field in %.









873

874

**Figure 4**: Temporal evolution of the ozone mass flux [Tg/year] from 1960 to 2099 in the transient RCP6.0 simulation integrated globally (black), over the northern (red), and southern (blue) hemispheres. The thin lines indicate the linear fits for the sub-periods 1960 to 2000 and 2000 to 2099. In addition, the ozone mass fluxes derived from the timeslice simulations (TS) for the years 2000 (REF2000) and 2100 (REF2100, RCP8.5 scenario) are shown by open circles including the ± 2σ range.

881





882

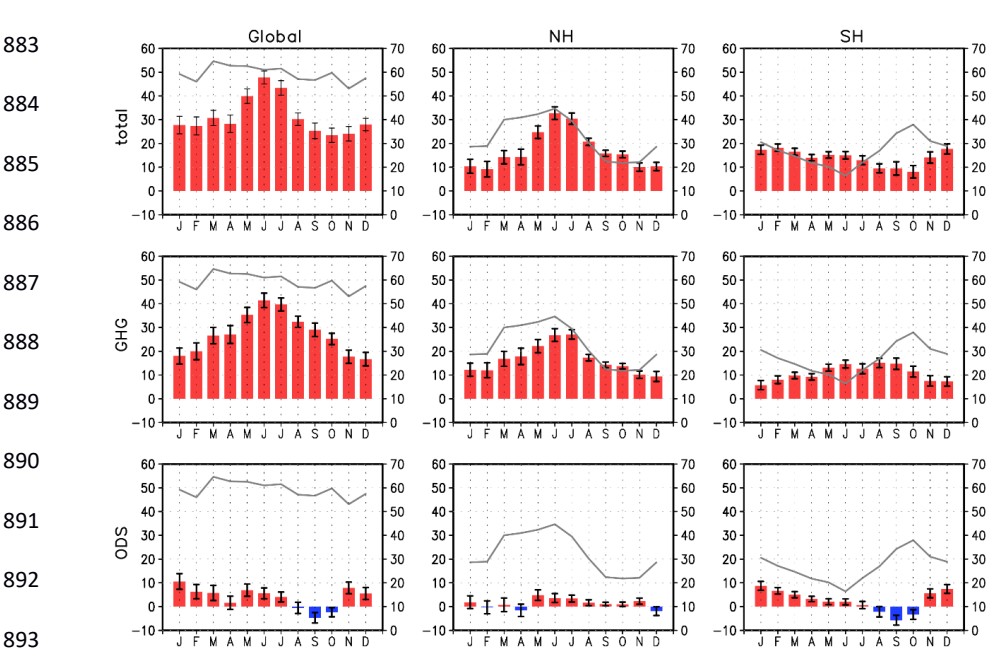

894

**Figure 5**: Annual cycle of ozone mass flux changes [Tg/month] in the timeslice simulations

integrated globally (left), over the NH (middle), and over the SH (right) for the changes due to

all forcings between 2000 and 2100 (top row), the effect of increasing GHG concentrations

(middle row) and the impact of declining ODS levels (bottom row). The black error bars denote

the $\pm 2\sigma$ standard deviation. The absolute ozone mass flux of the reference simulation REF2000

is shown as grey line with the corresponding y-axis on the right.














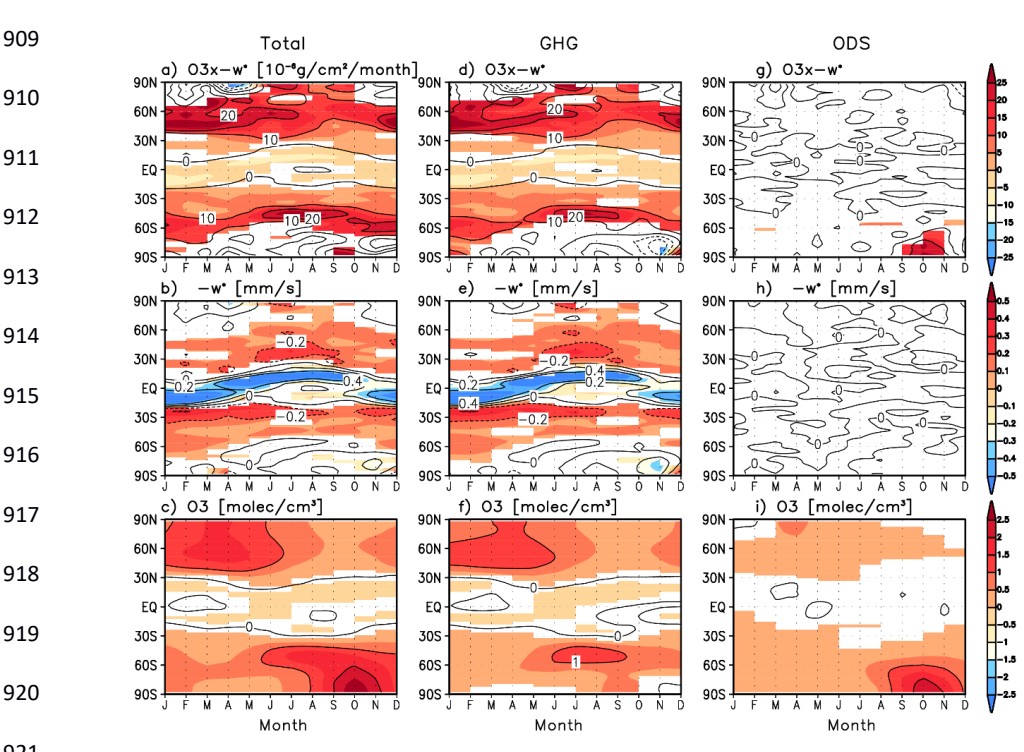

921

922

**Figure 6**: Annual cycle of the zonal mean change at 91 hPa in the timeslice simulations due to all forcings between 2000 and 2100 (left column), due to GHG increase (middle column), and due to ODS decrease (right column). Upper row: changes in the product of ozone concentration and $-\overline{w}^{*}$ [$10^{6}$ g/cm$^{2/}$month], which equals F$_{in}$ when integrated over all latitudes. Middle row: changes in $-\overline{w}^{*}$ [mm/s]. Bottom row: changes in the ozone concentration [molecules/cm$^{2}$]. Significant changes on the 95 % confidence level are colored.
























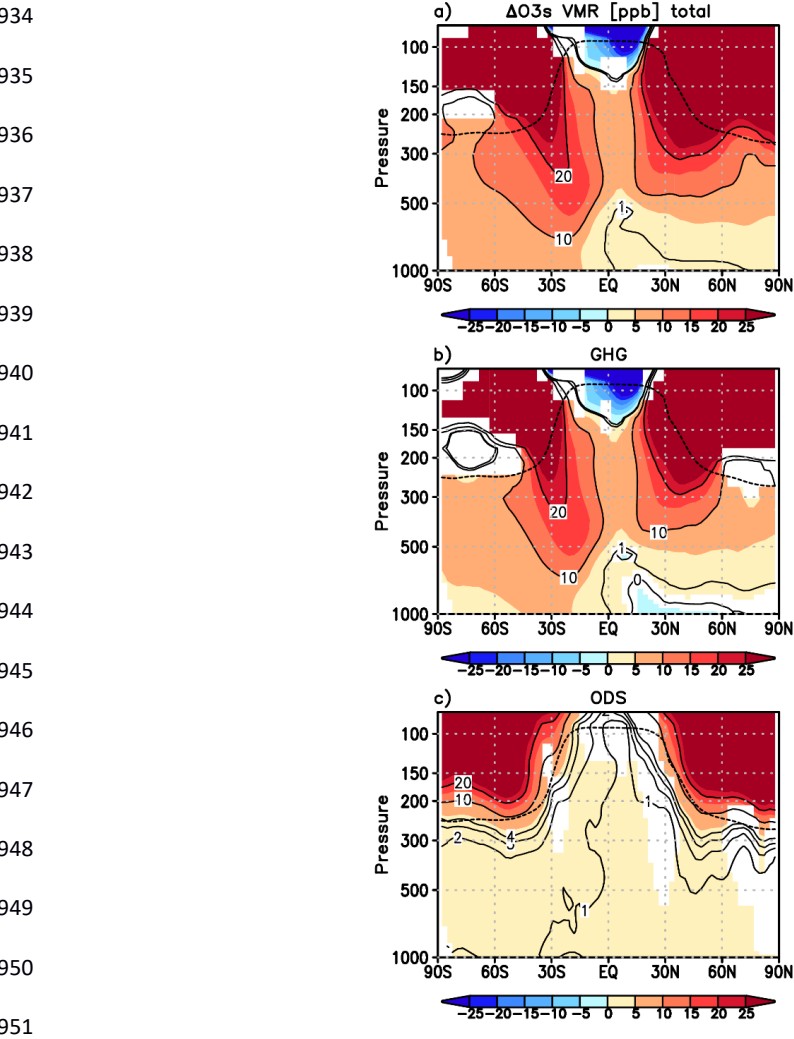

**Figure 7**: Changes in the volume mixing ratios [ppbv] of the diagnostic tracer O3s for a) the

changes between 2000 and 2100 due to all forcings, b) the changes between 2000 and 2100 due

to increasing GHG concentrations and c) the changes between 2000 and 2100 due to declining

ODS levels for June (when the ozone mass flux is maximum in the NH and minimum in the

SH; see Fig. 2a). Significant changes on the 95 %-confidence level are colored. The black dotted

line represents the mean tropopause position in the REF2000 simulation. For the small ODS-

induced changes (c) additional contour lines (2, 3, and 4 ppbv) are shown.





















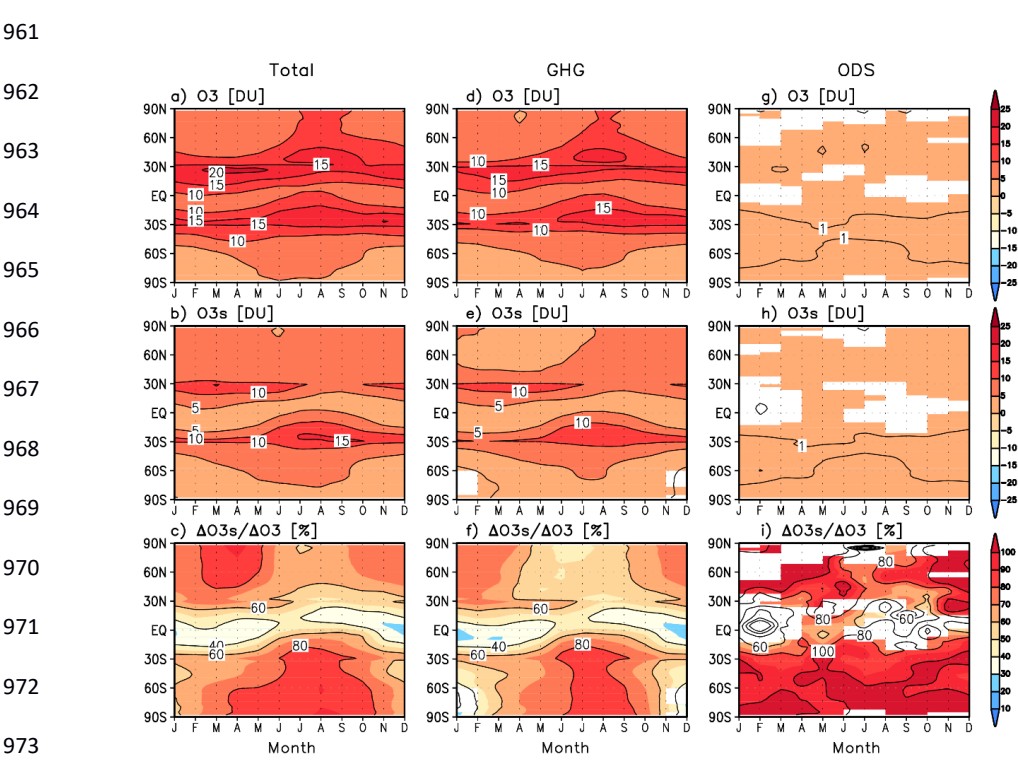

**Figure 8**: As Figure 6, but for the tropospheric ozone columns [DU] (upper row), the
tropospheric columns of O3s [DU] (middle row), and the contribution of the O3s changes to
the ozone changes between 2000 and 2100 [%] for the tropospheric column (bottom). Shading
in the bottom row indicates the regions where both, O3s and ozone changes, are significant on
the 95 % confidence level.



























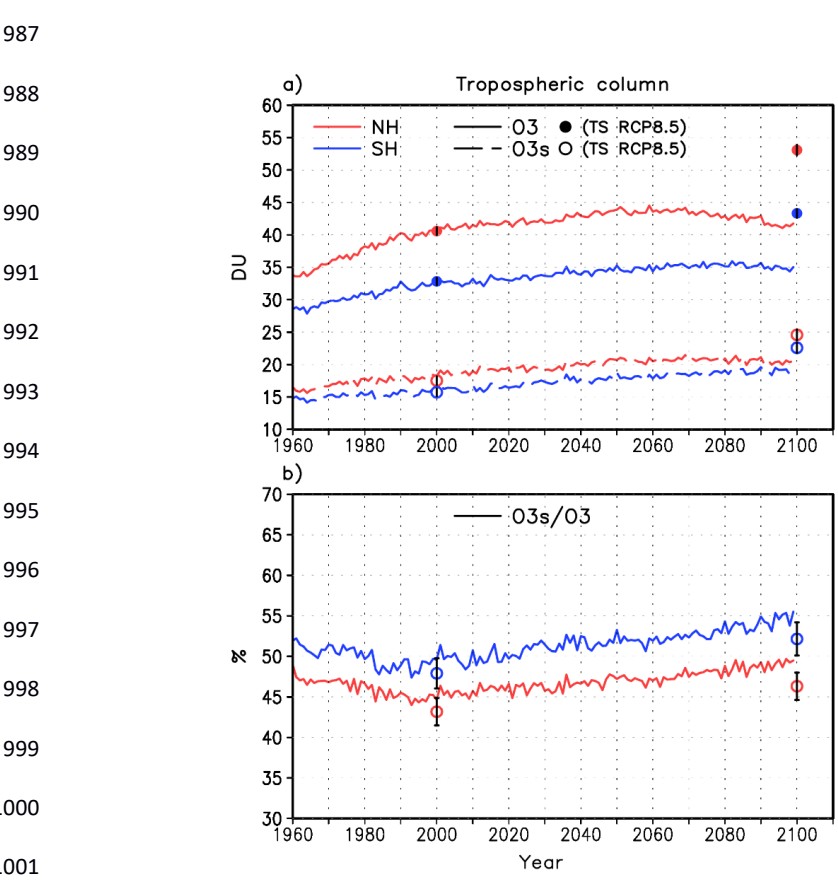

**Figure 9**: a) Temporal evolution of the annual mean tropospheric column in ozone (solid) and
O3s (dashed) [DU] averaged for the NH (red) and the SH (blue) in the transient RCP6.0
simulation and the corresponding values of the reference timeslice simulations for the year 2000
and 2100 (ozone: closed circle; O3s: open circle). b) Same as a) but for the ratio between O3s
and O3 [%]. The black bars denote the ± 2σ range for the timeslice simulations. Note that the
intra-annual variability in ozone and O3s is small in the timeslice simulations and the error bars
are very short.