# Peer review of "Future changes in the stratosphere-to-troposphere ozone mass flux and the contribution"

_Atmospheric Chemistry and Physics, 2018_

## Referee Comment (RC1) · Anonymous Referee #1 · 5 Mar 2018

(1) General comments:

Ozone in the troposphere is important for the radiative budget (i.e. greenhouse gas, GHG), atmospheric chemistry (e.g. main source of hydroxyl radicals) and air quality (i.e. pollutant with negative consequences for the biosphere). Thus it is crucial to understand its evolution during the 21st century under different pathways (i.e. emission scenarios). Indeed, a key term influencing the abundance and distribution of tropospheric ozone is the stratosphere-troposphere exchange (STE). The study investigates future changes of STE following the RCP8.5 and RCP6.0 emission scenarios based on experiments of the EMAC chemistry-climate model. Projections based on

chemistry-climate models are a valuable mean to probe ozone evolution associated with specific factors – i.e. such as ozone precursors, GHGs and ozone depleting substances (ODSs) – albeit significant disagreements compared to observations and between models are often found.

Overall, the study addresses relevant research questions with regard to the evolution of the STE, its drivers and the contribution of stratospheric-produced ozone to the tropospheric budget. The manuscript is technically well written. Specific comments are detailed below, which are intended to help the authors improve the paper. Briefly, I suggest more details about the "climates" (sea-surface temperatures and sea-ice concentrations, SSTs/SICs) imposed on the simulations to better interpret the comparison between the two emission scenarios considered, since they are from different models (e.g. different climate sensitivity) with consequences for ozone (e.g. chemistry and dynamics). Also, I suggest further acknowledgement to the various roles that methane plays on ozone chemistry – in the troposphere and the stratosphere – as well as its importance on the evolution of the STE. I understand in this set of simulations the effect of an enhanced Brewer-Dobson circulation and stratospheric cooling cannot be separated from the role of methane on ozone chemistry. The present study is recommended for publication after the specific and technical comments are addressed.

(2) Specific comments:

a. The study explores how two different emission scenarios (RCP6.0 and RCP8.5) affect the stratospheric ozone mass flux into the troposphere. There is, however, an inconsistency between the SSTs/SICs used that have not been explored/detailed. In order to compare and better understand the differences in STE between these scenarios, it would be desirable to discuss how the underlying Earth System Models (ESMs) – MPI-ESM and HadGEM2-ESM – project climate. For example, the difference in global surface warming by the end of the century (2081–2099 relative to 1985–2005) between these ESMs may be as large as ∼1 K (higher for HadGEM2-ESM) under the RCP8.5 (see Table 3 in Friedlingstein et al., 2014 – doi.org/10.1175/JCLI-D-12-00579.1). Moreover, Fig. 11 in Jöckel et al. (2016 – doi.org/10.5194/gmd-9-1153-2016) shows that internally generated (EMAC-MPIOM) global and annual mean SSTs for the same period but following the RCP6.0 scenario are ~1 K lower compared to those projected by HadGEM2-ES. Also, the CMIP5 ensemble projects that annual mean surface air temperature anomalies for the same period under the RCP6.0 and RCP8.5 are of $2.2 \pm 0.5$ K and $3.7 \pm 0.7$ K respectively (i.e. mean difference of 1.5 K; see Table 12.2 in IPCC, 2013). Therefore, the difference between the SSTs/SICs imposed on the simulations under these emission scenarios may be relatively small, implying that ozone precursor emissions (i.e. methane) may account for a large fraction of the STE difference (i.e. rather than climate-induced).

b. Methane is a GHG and plays various roles on ozone chemistry. Although, the authors already acknowledge the role of ozone precursors in the troposphere (i.e. methane; Page 5, lines 85–92; Page 20, lines 462–464), the effect on the stratosphere is not addressed (see below). Among other effects, it is an ozone precursor in the lower stratosphere (i.e. smog-like chemistry) and affects chlorine active/inactive partitioning, which is particularly important for polar regions under high concentrations of ODSs. An increase in methane abundances (e.g. boundary conditions) results in higher stratospheric ozone (e.g. Kawase et al., 2011 – doi.org/10.1029/2010GL046402; Revell et al., 2012 – doi.org/10.5194/acp-12-11309-2012), which will affect the stratospheric ozone mass flux into the troposphere. I guess in the GHG2100 simulation, methane is coupled to both, radiation and chemistry schemes. Therefore, the climate-induced and methane-related impacts on the STE cannot be unambiguously separated (i.e. both included in the "GHG-induced changes"). I think this should be addressed more clearly in the manuscript and its main findings.

(3) Technical comments:

Pages 1–2, lines 24–25. "... GHG effect on the STE change is due to circulation and stratospheric ozone changes,...". Does the latter refer to stratospheric cooling and/or methane-related chemistry? Please clarify.

[Figure]

Pagea 1–2, Abstract. Research question (4) refers to a comparison between the emission scenarios, yet I find no reference to this in the abstract.

Page 4, lines 77–79. Note Banerjee et al. (2016) in their "climate" simulations did not include methane (and N2O) in the chemistry scheme.

Page 6, lines 111–113. The sentence reads a bit confusing – i.e. 1960–1999 period is not RCP6.0.

Page 7, lines 150–152. I guess "well-mixed GHGs" are coupled to the radiation and chemistry schemes. Please clarify, this is important to interpret the findings.

Page 11, line 244. Typo: "... Eqation".

Page 11, line 246. "... ozone flux into the NH is larger than into the SH..." Is this significant?

Page 12, lines 252–259. Here it would be appropriate to include the STE budget term from the ACCMIP ensemble (i.e. 477 ± 96 Tg/year; mean and sdev) that informed the last (AR5) IPCC assessment report (see Table 2 in Young et al., 2013; doi.org/10.5194/acp-13-2063-2013).

Page 12, lines 259–261. Note Olsen et al. (2002) presented an influx estimate for the NH midlatitudes (e.g. see their Table 1 for further estimates).

Page 13, line 289. Do you mean "O3s tropospheric columns..."?

Page 14, lines 302–303. "low(er)... small(er)"?

Page 14, lines 309–311. I would also include where your model lies compared to the last IPCC report. As the authors commented, there is still a relatively large uncertainty on the STE term.

Page 15, lines 325–326. Could the "... unintended neglect of minor chlorine source gases..." account partly for the difference on ozone mass flux trends compared to

[Figure]

Hegglin and Shepherd (2009)?

Page 15, line 346. I would explain a bit more what the "... more extreme GHG emission scenario" means (i.e. climate-induced and methane-related impacts).

Page 16, lines 356–359. From Fig. 5 it seems that non-linearities may not have a significant impact on the STE, and that ODS-only and GHG-only largely account for the total change (i.e. largest changes in ozone precursors are for methane).

Page 17, lines 385–389; Page 19, lines 419–420; Page 20, lines 460–462; Page 23, lines 522–524. I think the effect of methane-related chemistry on the STE and ozone should be included (assuming methane is coupled to the radiation and chemistry schemes).

Page 18, lines 403–405. Please clarify this sentence.

Page 20, lines 456–457. This is a robust result from different CCMs and worth mentioning (Banerjee et al., 2017 – doi.org/10.5194/acp-2017-741; Iglesias–Suarez et al., 2017 – doi.org/10.5194/acp-2017-939).

Page 41 and 46, Figure 4 and 9. "TS RCP8.5" legend is a bit confusing since it includes year 2000 (i.e. REF2000).

Pages 43–44, Figure 6 and 7. How is the 95 % confidence level calculated? Please explain.

Page 45, Figure 8. It would be very helpful to have the annual and global mean values [DU] and [%] at the top of the figures.

---

## Referee Comment (RC2) · Anonymous Referee #2 · 7 Mar 2018

General assessment

This manuscript investigates the role of stratosphere-troposphere exchange of ozone on the tropospheric ozone budget and its drivers – climate change and declining ODS concentrations – over the 21st century. In general, the evaluations are straightforward and the manuscript is well written. While I see some shortcomings in the experimental setup of the simulations to answer the questions posed in a clean way (see major and minor comments in the following), the study offers some new and interesting insights, which warrant publication.

Major comments

[Figure]

A major issue of this study results from trying to answer question 5 – How is the ratio of stratospheric ozone in the troposphere changing in the future? The problem lies in how the authors have set up their simulations to answer the questions 1-5 posed, and that they do not allow for a clean separation of the different factors (ozone precursor, ODS, CH4, GHG effects) influencing stratosphere-troposphere exchange of ozone. As such the quantification of the relative contributions of the climate and ODS drivers seems somewhat unsatisfying, since the ratio of stratospheric to total ozone in the troposphere depends also on the mixture of the other contributing factors driving ozone in the simulations. While the study still is insightful, the shortcomings (contributions of CH4 and ozone precursor induced ozone production) need to be discussed more thoroughly.

If I understand right, Figure 8 is meant to quantify the relative contributions of ODS-decline and GHG-increases to the stratospheric ozone contribution to tropospheric ozone increases. However, I think the discussion of this Figure is not very logical. On P20 you focus first on the discussion of the relative contributions, which doesn't answer the question you pose at the beginning of the paragraph and in my eyes puts the wrong emphasis on these results. Per definition or design of your simulations, it is a given that the stratospheric contribution in the ODS-only simulation will be the main effect on tropospheric ozone levels (since nothing else is changing), so nothing surprising. The smallish regional differences can be explained by inconsistencies in boundary condition settings or internal variability of the climate system. Hence, the absolute numbers and their relative contributions to the ozone changes in the full simulation are of much more relevance and should be discussed first.

List of comments

Abstract: I feel that the abstract could be much improved by summarising more succinctly and more clearly the main conclusions of the manuscript. At the moment they read like a conclusions section with too much (and nevertheless not enough) detail to me. In particular L29-30 are unclear to me, since you do not explain through which

mechanism GHGs lead to increased tropospheric ozone loss.

Abstract L31: The notation of 'stratospheric column ozone in the troposphere' is confusing to me and seems too close to the notion of the stratospheric column ozone we usually refer to (that resides in the stratosphere). Could you say 'the column-integrated O3s in the troposphere' instead, or anything similar?

P8L178ff: Appenzeller et al (1996) did only address the mass flux, not the ozone flux with their approach. It is important to note that the approach you follow is that of Hegglin and Shepherd (2009), which has to be seen as an extension of Appenzeller et al. Please add this reference to reflect this.

P4L77 Usually, scientists refer to 'idealized model simulations' where simplified models are used and not full-blown chemistry-climate models that hopefully are at least somewhat realistic. Suggest rephrasing here and further down (L104) as well.

P6L188 correct to '…change in the future.'

P7L150 Using only 5 years of spin-up seem somewhat short to me given that you would need to bring the stratosphere (with transport times around 5 years in the upper stratosphere) into an updated state. Did you make any tests to see whether the model has no remaining drifts?

P10L209-215 The explanation of this alternative method of estimating STE is unclear to me. Did these references really use the loss of O3s and not O3 to infer STE as a residual from O3 production/loss? What additional information would this yield compared to looking at O3s change as you do here?

P10L223ff and P11L232 It would have been more convincing to compare the model results here to actual measurements in these figures to test the realism of the transition region and stratospheric transport in EMAC. However, I realize that this may be beyond the scope of this paper and do hence not request you to do so. However, it is difficult to say what you learn from a comparison with assimilated MOPPITT data as shown

in Barre et al (2013), since these data do not have the required vertical resolution to resolve the transition region. I suggest to instead compare to the ACE-FTS derived correlations in Hegglin et al. (2009) who conveniently show the CO-O3 correlations for the 30-60N latitude band in DJF (and other seasons, see their figure 7) as you have chosen in your figure. Here you see (in contrast to the Tian et al (2010) paper) that the CO-O3 correlations has a strong seasonality and latitudinal dependency. Judging by eye in the apple-to-apple comparison when using the Hegglin et al FIgure, I would say EMAC is resolving the transition very well, not just reasonably well with CO values at O3=0.1, 0.5, 1,5 ppmv of around 90, 30, 17 ppbv, respectively.

Hegglin, M. I., C. D. Boone, G. L. Manney, K. A. Walker, A global view of the extra-tropical tropopause transition layer from Atmospheric Chemistry Experiment Fourier Transform Spectrometer O3, H2O, and CO, J. Geophys. Res., 114, D00B11, doi:10.1029/2008JD009984, 2009.

P11L228 The reference to Pan et al 2007 seems missing.

P12L259 IPCC (2013) has a newer compilation/assessment of STE ozone fluxes de-rived from different methodologies, please update to the range indicated there.

P13L276-8 It seems to me that the ozone influx from the stratosphere is larger in spring than in summer according to your results in Figure 2.

P14L280 correct to '…in the SH.'

P14L282 again, I would prefer here '…the column-aggregated stratospheric ozone in the troposphere…'

P14L311 Please update this statement with respect to the IPCC 2013 results.

Tables 2 and 3: Please provide statistical uncertainties for your trend estimates.

P16L358 See also major comment above. It seems crucial to highlight already in the methodology section that precursor emissions are not evolving in the GHG- and ODS-

only simulations. Or did I miss this point? Can you provide an argument/estimate of the effect changes in precursor emissions could have on your results? This seems a major inconsistency in the design of your study with a potentially large effect on the amounts of ozone with stratospheric-origin in the troposphere that should be more thoroughly discussed also in the results and conclusion section. Not only are ozone precursor emissions potentially affecting the lifetime of O3s in the troposphere, but models predict that they had a major effect on lowermost stratospheric ozone concentrations as well, which will likely influence your derived ozone fluxes through non-linear chemical reactions.

P17L372-5 Reading your manuscript, I found this to be a very interesting and puzzling result, which you explain further down in more detail. However, to better envision what is going on I would appreciate to see how -dM/dt changes over time in particular given that this is the second major term in determining the ozone flux into the troposphere. I hence suggest adding a figure that quantifies the LMS mass changes over the 21st century.

P19L424-6 I do not see that O3s is transported further down in the NH than in the SH in Figure 7, rather it seems the opposite. Also, the explanation seems not really an explanation since the chemical lifetime is equally long in the SH winter than in the NH winter. Please check.

P20L447 correct to '... a similarly strong ...'

Figure 8 caption: Please indicate that the numbers you show are changes in ozone and not absolute amounts and do not just refer to Figure 6. Using delta O3 in the figure titles would achieve the same result.

Figure 9 caption L1008: did you mean 'inter-annual'?

P21L484 Sentence seems incomplete.

P21L386-8 Another factor that needs to be discussed are tropospheric ozone precursor

emissions and their effect on the tropospheric ozone burden in the RCP6.0 simulation, since this will be another confounding factor when discussing relative changes in O3s contributions to total tropospheric ozone.

P22L504 correct to '. . . will consist of 46% ozone from. . . '

---

## Author Comment (AC1) · 30 Apr 2018

Reply to Anonymous Referee #1

We would like to thank the referee for the time and the useful comments that helped to clarify important aspects of the manuscript. Our replies to the comments have been added in blue.

(1) General comments:

Ozone in the troposphere is important for the radiative budget (i.e. greenhouse gas, GHG), atmospheric chemistry (e.g. main source of hydroxyl radicals) and air quality (i.e. pollutant with negative consequences for the biosphere). Thus it is crucial to understand its evolution during the 21st century under different pathways (i.e. emission scenarios). Indeed, a key term influencing the abundance and distribution of tropospheric ozone is the stratosphere-troposphere exchange (STE). The study investigates future changes of STE following the RCP8.5 and RCP6.0 emission scenarios based on experiments of the EMAC chemistry-climate model. Projections based on chemistry-climate models are a valuable mean to probe ozone evolution associated with specific factors – i.e. such as ozone precursors, GHGs and ozone depleting substances (ODSs) – albeit significant disagreements compared to observations and between models are often found.

Overall, the study addresses relevant research questions with regard to the evolution of the STE, its drivers and the contribution of stratospheric-produced ozone to the tropospheric budget. The manuscript is technically well written. Specific comments are detailed below, which are intended to help the authors improve the paper. Briefly, I suggest more details about the "climates" (sea-surface temperatures and sea-ice concentrations, SSTs/SICs) imposed on the simulations to better interpret the comparison between the two emission scenarios considered, since they are from different models (e.g. different climate sensitivity) with consequences for ozone (e.g. chemistry and dynamics). Also, I suggest further acknowledgement to the various roles that methane plays on ozone chemistry – in the troposphere and the stratosphere – as well as its importance on the evolution of the STE. I understand in this set of simulations the effect of an enhanced Brewer-Dobson circulation and stratospheric cooling cannot be separated from the role of methane on ozone chemistry. The present study is recommended for publication after the specific and technical comments are addressed.

We thank the referee for addressing these two important aspects. We have addressed both in detail in our responses to the specific comments below, and adapted the manuscript accordingly.

(2) Specific comments:

a. The study explores how two different emission scenarios (RCP6.0 and RCP8.5) affect the stratospheric ozone mass flux into the troposphere. There is, however, an inconsistency between the SSTs/SICs used that have not been explored/detailed. In order to compare and better understand the differences in STE between these scenarios, it would be desirable to discuss how the underlying Earth System Models (ESMs) – MPI-ESM and HadGEM2-ESM – project climate. For example, the difference in global surface warming by the end of the century (2081–2099 relative to 1985–2005) between these ESMs may be as large as 1 K (higher for HadGEM2-ESM) under the RCP8.5 (see Table 3 in Friedlingstein et al., 2014 – doi.org/10.1175/JCLI-D-12-00579.1). Moreover, Fig. 11 in Jöckel et al. (2016 – doi.org/10.5194/gmd-9-1153-2016) shows that internally generated (EMAC-MPIOM) global

and annual mean SSTs for the same period but following the RCP6.0 scenario are 1 K lower compared to those projected by HadGEM2-ES. Also, the CMIP5 ensemble projects that annual mean surface air temperature anomalies for the same period under the RCP6.0 and RCP8.5 are of 2.2 ± 0.5 K and 3.7 ± 0.7 K respectively (i.e. mean difference of 1.5 K; see Table 12.2 in IPCC, 2013). Therefore, the difference between the SSTs/SICs imposed on the simulations under these emission scenarios may be relatively small, implying that ozone precursor emissions (i.e. methane) may account for a large fraction of the STE difference (i.e. rather than climate-induced).

We thank the referee for this important comment. We have addressed this issue in the revision by including the following text in Section 4:

*"Note that the difference in OMF change by 2100 between the timeslice and transient simulations may not be due to the different GHG scenario alone. The HadGEM2-ES model which provided the SST/SIC distribution for the transient RCP6.0 simulation is known to have a higher climate sensitivity than the MPI-ESM which provided the SST/SICs for the EMAC timeslice runs (Andrews et al., 2012). This might lead to a somewhat stronger future SST increase, stratosphere-troposphere exchange and OMF in the transient RCP6.0 run than would arise using MPI-ESM SSTs/SICs. Hence, the differences in OMF by 2100 discussed here represent a lower boundary estimate of the expected OMF differences between the RCP6.0 and RCP8.5 scenarios."*

However, the following figure shows that the MPI-ESM RCP8.5 SSTs are clearly higher than the HadGEM2-ES RCP6.0 SSTs by the end of the century.

[Figure]

Figure 1: Annual mean SSTs from 1960 to 2100 averaged over 20°S-20°N for RCP4.5 (red) and RCP 8.5 (blue) GHG scenarios derived from MPI-ES, and for RCP6.0 derived from HadGEM2-ES (green). The RCP8.5 timeslice simulations in this study used 2095-2104 averaged SSTs from RCP8.5; the transient RCP6.0 used the transient RCP6.0 SSTs (green curve).

b. Methane is a GHG and plays various roles on ozone chemistry. Although, the authors already acknowledge the role of ozone precursors in the troposphere (i.e. methane; Page 5, lines 85–92; Page 20, lines 462–464), the effect on the stratosphere is not addressed (see below). Among other effects, it is an ozone precursor in the lower stratosphere (i.e. smog-like chemistry) and affects chlorine active/inactive partitioning, which is particularly important for polar regions under high concentrations of ODSs. An increase in methane abundances (e.g. boundary conditions) results in higher stratospheric ozone (e.g. Kawase et al., 2011 – doi.org/10.1029/2010GL046402; Revell et al., 2012 – doi.org/10.5194/acp-12-11309-2012), which will affect the stratospheric ozone mass flux into the troposphere. I guess in the GHG2100 simulation, methane is coupled to both, radiation and chemistry schemes. Therefore, the climate-induced and methane-related impacts on the STE cannot

be unambiguously separated (i.e. both included in the "GHG-induced changes"). I think this should be addressed more clearly in the manuscript and its main findings.

We included the following section to the introduction:

"*Besides the temperature effect of the GHGs (mainly $CO_2$) on the ozone chemistry, the increasing abundances of $CH_4$ and $N_2O$ also have an impact on the net production of stratospheric ozone: While higher $N_2O$ concentrations are associated with an enhanced ozone loss in the stratosphere due to reactive nitrogen, a $CH_4$ increase not only causes a larger ozone loss in the lower and upper stratosphere but also leads to an increased ozone production in the lower stratosphere where it acts as an ozone precursor (e.g. Revell et al., 2012). In addition, $CH_4$ plays a role in polar ozone chemistry since changing $CH_4$ concentrations also alter the partitioning between halogen reservoir gases and activated halogen species (e.g. Nevison et al., 1999). The combined effect of the increasing GHG concentrations (including the interactions between the chemical cycles) on the net stratospheric ozone production is positive (e.g. Meul et al., 2014).*"

We agree that in our simulations the climate-induced and the methane related impacts of GHG changes on STE cannot be separated, but this was not our intention but instead to distinguish between the GHG and ODS related effects on STE and tropospheric ozone change. A comment has been added in Section 2.1 for clarification.

(3) Technical comments:

Pages 1–2, lines 24–25. ". . . GHG effect on the STE change is due to circulation and stratospheric ozone changes,. . .". Does the latter refer to stratospheric cooling and/or methane-related chemistry? Please clarify.

The abstract has been rewritten to improve clarity.

Pagea 1–2, Abstract. Research question (4) refers to a comparison between the emission scenarios, yet I find no reference to this in the abstract.

The abstract has been rewritten to improve clarity.

Page 4, lines 77–79. Note Banerjee et al. (2016) in their "climate" simulations did not include methane (and N2O) in the chemistry scheme.

A respective comment has been added to the text.

Page 6, lines 111–113. The sentence reads a bit confusing – i.e. 1960–1999 period is not RCP6.0.

The sentence has been rewritten.

Page 7, lines 150–152. I guess "well-mixed GHGs" are coupled to the radiation and chemistry schemes. Please clarify, this is important to interpret the findings.

This has been clarified in the text.

Page 11, line 244. Typo: ". . . Eqation".

Corrected

Page 11, line 246. ". . . ozone flux into the NH is larger than into the SH. . ." Is this significant?

The difference between NH maximum and SH maximum is not significant so we removed this statement from the manuscript.

Page 12, lines 252–259. Here it would be appropriate to include the STE budget term from the ACCMIP ensemble (i.e. 477 ± 96 Tg/year; mean and sdev) that informed the last (AR5) IPCC assessment report (see Table 2 in Young et al., 2013; doi.org/10.5194/acp-13-2063-2013).

The paragraph has been rewritten including information from ACCENT, ACCMIP and IPCC (2013).

Page 12, lines 259–261. Note Olsen et al. (2002) presented an influx estimate for the NH midlatitudes (e.g. see their Table 1 for further estimates).

Thank you for this note. We now use the values from IPCC (2013) based on Young et al. (2013).

Page 13, line 289. Do you mean "O3s tropospheric columns. . ."?

Yes, 'tropospheric' added

Page 14, lines 302–303. "low(er). . . small(er)"?

Yes, corrected

Page 14, lines 309–311. I would also include where your model lies compared to the last IPCC report. As the authors commented, there is still a relatively large uncertainty on the STE term.

A broader discussion of the STE analyses from IPCC (2013) and model intercomparisons has been included in Section 3.2

Page 15, lines 325–326. Could the ". . . unintended neglect of minor chlorine source gases. . ." account partly for the difference on ozone mass flux trends compared to Hegglin and Shepherd (2009)?

Yes, the weaker ozone depletion due to the missing minor chlorine gases could contribute to the smaller negative trend. But we don't have the possibility to quantify this effect, so decided not to discuss this in the text.

Page 15, line 346. I would explain a bit more what the ". . . more extreme GHG emission scenario" means (i.e. climate-induced and methane-related impacts).

'climate change- and methane-related effects' has been included in the text.

 Page 16, lines 356–359. From Fig. 5 it seems that non-linearities may not have a significant impact on the STE, and that ODS-only and GHG-only largely account for the total change (i.e. largest changes in ozone precursors are for methane).

Unfortunately, we are not able to test the impact of non-linearities but we agree that the effect is probably small.

Page 17, lines 385–389; Page 19, lines 419–420; Page 20, lines 460–462; Page 23, lines 522–524. I think the effect of methane-related chemistry on the STE and ozone should be included (assuming methane is coupled to the radiation and chemistry schemes).

We have rewritten the statement:

"*The increase of Fin (or O3 x -$\overline{w}^*$) due to the GHG effect (Fig. 6d) is caused by an increase in the downwelling (i.e. -$\overline{w}^*$, positive for downwelling, Fig. 6e) of the BDC in the winter season with climate*

*change (e.g., Sudo et al., 2003; Butchart et al., 2010; Oberländer et al., 2013) in combination with an ozone increase resulting from modified chemical production and loss rates in the stratosphere and enhanced meridional transport (Figure 6f)."*

Page 18, lines 403–405. Please clarify this sentence.

We have added a new figure (Figure 7) showing the changes in the seasonal breathing term. We think that this make this section clearer.

Page 20, lines 456–457. This is a robust result from different CCMs and worth mentioning (Banerjee et al., 2017 – doi.org/10.5194/acp-2017-741; Iglesias–Suarez et al., 2017 – doi.org/10.5194/acp-2017-939).

Both manuscripts address the radiative forcing due to changes in ozone which is not a topic of this manuscript. Moreover, Iglesias-Suarez et al. (2017) is still under review. Therefore, the references have not been added.

Page 41 and 46, Figure 4 and 9. "TS RCP8.5" legend is a bit confusing since it includes year 2000 (i.e. REF2000).

We have modified the Figures. Now it is stated "TS obs+RCP8.5".

Pages 43–44, Figure 6 and 7. How is the 95 % confidence level calculated? Please explain.

The statistical significance of the future changes is tested by applying the Student's t-test. We have included this information in the text.

Page 45, Figure 8. It would be very helpful to have the annual and global mean values [DU] and [%] at the top of the figures.

Done

---

## Author Comment (AC2) · 30 Apr 2018

Reply to Anonymous Referee #2

We would like to thank the referee for the time and the useful comments that helped to clarify important aspects of the manuscript. Our replies to the comments have been added in blue.

General assessment

This manuscript investigates the role of stratosphere-troposphere exchange of ozone on the tropospheric ozone budget and its drivers – climate change and declining ODS concentrations – over the 21st century. In general, the evaluations are straightforward and the manuscript is well written. While I see some shortcomings in the experimental setup of the simulations to answer the questions posed in a clean way (see major and minor comments in the following), the study offers some new and interesting insights, which warrant publication.

Major comments

A major issue of this study results from trying to answer question 5 – How is the ratio of stratospheric ozone in the troposphere changing in the future? The problem lies in how the authors have set up their simulations to answer the questions 1-5 posed, and that they do not allow for a clean separation of the different factors (ozone precursor, ODS, CH4, GHG effects) influencing stratosphere-troposphere exchange of ozone. As such the quantification of the relative contributions of the climate and ODS drivers seems somewhat unsatisfying, since the ratio of stratospheric to total ozone in the troposphere depends also on the mixture of the other contributing factors driving ozone in the simulations. While the study still is insightful, the shortcomings (contributions of CH4 and ozone precursor induced ozone production) need to be discussed more thoroughly.

The major goal of our study was to investigate the impacts of future changes in both GHG increases and ODS decline between 2000 and 2100 on STE and tropospheric ozone, both for the combined effect and the separate GHG and ODS effects. The additional effect of changes in tropospheric ozone production by changes in ozone precursors was implicitly included in the simulations: In the REF2100 simulation all ozone precursors were adapted to 2100; in the GHG2100 simulation the $CH_4$ GHG precursor was increased to 2100 values. With this model setup the goals of our study could be addressed. To clarify the inclusion of CH4 and precursors text has been added in Sections 2 and 4.

If I understand right, Figure 8 is meant to quantify the relative contributions of ODS decline and GHG-increases to the stratospheric ozone contribution to tropospheric ozone increases. However, I think the discussion of this Figure is not very logical. On P20 you focus first on the discussion of the relative contributions, which doesn't answer the question you pose at the beginning of the paragraph and in my eyes puts the wrong emphasis on these results. Per definition or design of your simulations, it is a given that the stratospheric contribution in the ODS-only simulation will be the main effect on tropospheric ozone levels (since nothing else is changing), so nothing surprising. The smallish regional differences can be explained by inconsistencies in boundary condition settings or internal variability of the climate system. Hence, the absolute numbers and their relative contributions to the ozone changes in the full simulation are of much more relevance and should be discussed first.

We are sorry but are not able to understand completely the referee's comment. In Fig. 9 (old Fig. 8) the left column shows from top to bottom the absolute change in the tropospheric ozone column, the absolute change in the tropospheric column of stratospheric ozone, and how much of tropospheric ozone column change is due by stratospheric ozone (in %), all that for the total forcing

by GHG and ODS changes. The middle column shows the same quantities only for GHG increase; the right column for ODS decline. We agree that in the GHG and ODS cases, change patterns are only due to the implied forcing change (and nothing else). And it is true that in the ODS run, the relative contribution from the stratosphere is very high because there is no CH4 increase in the troposphere (hence no changed ozone production) and the effects of dynamical changes are minor (although the stratospheric contribution is not 100% everywhere). But the benefit of this figure – in our opinion - is that by comparing the change patterns of the individual forcings with those by the total forcing allows us to derive which forcing affects which region to which extent.

List of comments

Abstract: I feel that the abstract could be much improved by summarising more succinctly and more clearly the main conclusions of the manuscript. At the moment they read like a conclusions section with too much (and nevertheless not enough) detail to me. In particular L29-30 are unclear to me, since you do not explain through which mechanism GHGs lead to increased tropospheric ozone loss.

The abstract has been rewritten and hopefully clarified.

Abstract L31: The notation of 'stratospheric column ozone in the troposphere' is confusing to me and seems too close to the notion of the stratospheric column ozone we usually refer to (that resides in the stratosphere). Could you say 'the column-integrated O3s in the troposphere' instead, or anything similar?

We have changed the notation.

P8L178ff: Appenzeller et al (1996) did only address the mass flux, not the ozone flux with their approach. It is important to note that the approach you follow is that of Hegglin and Shepherd (2009), which has to be seen as an extension of Appenzeller et al. Please add this reference to reflect this.

Reference Hegglin and Shepherd (2009) has been added to the text and the method clarified.

P4L77 Usually, scientists refer to 'idealized model simulations' where simplified models are used and not full-blown chemistry-climate models that hopefully are at least somewhat realistic. Suggest rephrasing here and further down (L104) as well.

'idealized' not really needed here and removed at both places

P6L188 correct to '. . .change in the future.'

done

P7L150 Using only 5 years of spin-up seem somewhat short to me given that you would need to bring the stratosphere (with transport times around 5 years in the upper stratosphere) into an updated state. Did you make any tests to see whether the model has no remaining drifts?

Yes, we checked that there were no drifts in the analysis periods of the model runs. The figure shows as an example global annual mean temperature of the REF2000 run (the analysis period extends from model year 2005 to model year 2045; all model years are run with year 2000 conditions).

[Figure]

P10L209-215 The explanation of this alternative method of estimating STE is unclear to me. Did these references really use the loss of O3s and not O3 to infer STE as a residual from O3 production/loss? What additional information would this yield compared to looking at O3s change as you do here?

Yes, they use the loss of the diagnostic O3s tracer. However, since this is not relevant here we have removed this part.

P10L223ff and P11L232 It would have been more convincing to compare the model results here to actual measurements in these figures to test the realism of the transition region and stratospheric transport in EMAC. However, I realize that this may be beyond the scope of this paper and do hence not request you to do so. However, it is difficult to say what you learn from a comparison with assimilated MOPPITT data as shown in Barre et al (2013), since these data do not have the required vertical resolution to resolve the transition region. I suggest to instead compare to the ACE-FTS derived correlations in Hegglin et al. (2009) who conveniently show the CO-O3 correlations for the 30-60N latitude band in DJF (and other seasons, see their figure 7) as you have chosen in your figure. Here you see (in contrast to the Tian et al (2010) paper) that the CO-O3 correlations has a strong seasonality and latitudinal dependency. Judging by eye in the apple-to-apple comparison when using the Hegglin et al FIgure, I would say EMAC is resolving the transition very well, not just reasonably well with CO values at O3=0.1, 0.5, 1,5 ppmv of around 90, 30, 17 ppbv, respectively.

The reference to the MOPPITT data has been removed and a comparison to ACE-FTS in Hegglin et al. (2009) included in the text instead. 'reasonably' has been removed.

Hegglin, M. I., C. D. Boone, G. L. Manney, K. A. Walker, A global view of the extratropical tropopause transition layer from Atmospheric Chemistry Experiment Fourier Transform Spectrometer O3, H2O, and CO, J. Geophys. Res., 114, D00B11, doi:10.1029/2008JD009984, 2009.

Reference has been added.

P11L228 The reference to Pan et al 2007 seems missing.

Pan et al. (2007) has been added in the list of references.

P12L259 IPCC (2013) has a newer compilation/assessment of STE ozone fluxes derived from different methodologies, please update to the range indicated there.

STE ozone fluxes from Table 8.1 in IPCC (2013) are used for comparison now. The section has been rewritten accordingly.

P13L276-8 It seems to me that the ozone influx from the stratosphere is larger in spring than in summer according to your results in Figure 2.

This is true for the SH, but in the NH the largest influx occurs in early summer. Text has been clarified.

P14L280 correct to '. . .in the SH.'

done

P14L282 again, I would prefer here '. . .the column-aggregated stratospheric ozone in the troposphere. . .'

Has been added.

P14L311 Please update this statement with respect to the IPCC 2013 results.

The reference to IPCC (2001) has been removed.

Tables 2 and 3: Please provide statistical uncertainties for your trend estimates.

The uncertainty ranges were included in Table 2 and 3. It has been added that all changes are significant on the 95% confidence level.

P16L358 See also major comment above. It seems crucial to highlight already in the methodology section that precursor emissions are not evolving in the GHG- and ODSC4 only simulations. Or did I miss this point? Can you provide an argument/estimate of the effect changes in precursor emissions could have on your results? This seems a major inconsistency in the design of your study with a potentially large effect on the amounts of ozone with stratospheric-origin in the troposphere that should be more thoroughly discussed also in the results and conclusion section. Not only are ozone precursor emissions potentially affecting the lifetime of O3s in the troposphere, but models predict that they had a major effect on lowermost stratospheric ozone concentrations as well, which will likely influence your derived ozone fluxes through non-linear chemical reactions.

The reference simulation for the year 2100 includes changes in tropospheric precursor emissions, hence future changes in tropospheric ozone production (in addition to changes in STE). In addition, the GHG2100 simulation considering the effect of increased GHG concentrations includes increases in $CH_4$, i.e. the ozone precursor with the strongest trend in the RCP8.5 scenario, and consequently changes in tropospheric ozone production.

We have clarified this now by adding text in Sections 2 and 4.

P17L372-5 Reading your manuscript, I found this to be a very interesting and puzzling result, which you explain further down in more detail. However, to better envision what is going on I would appreciate to see how -dM/dt changes over time in particular given that this is the second major term in determining the ozone flux into the troposphere. I hence suggest adding a figure that quantifies the LMS mass changes over the 21st century.

A new Figure 7 has been added showing dM/dt in the same format as Fig. 5. The text has been expanded to explain the seasonal variation of the seasonal breathing term for the total, GHG and ODS forcings.

P19L424-6 I do not see that O3s is transported further down in the NH than in the SH in Figure 7, rather it seems the opposite. Also, the explanation seems not really an explanation since the chemical lifetime is equally long in the SH winter than in the NH winter. Please check.

The text states: "In the SH, the abundance of stratospheric ozone increases throughout the troposphere down to the surface. More O3s seems to be transported further down than in the NH,…". "Here" has been included to avoid misunderstanding.

P20L447 correct to '. . . a similarly strong . . .'

Done

Figure 8 caption: Please indicate that the numbers you show are changes in ozone and not absolute amounts and do not just refer to Figure 6. Using delta O3 in the figure titles would achieve the same result.

Done

Figure 9 caption L1008: did you mean 'inter-annual'?

Yes; changed.

P21L484 Sentence seems incomplete.

We removed the sentence.

P21L386-8 Another factor that needs to be discussed are tropospheric ozone precursor emissions and their effect on the tropospheric ozone burden in the RCP6.0 simulation, since this will be another confounding factor when discussing relative changes in O3s contributions to total tropospheric ozone.

The role of tropospheric ozone precursors in the RCP6.0 and RCP8.5 simulations on the relative changes in O3s contributions in the troposphere had already been discussed in the manuscript by the following sentences:

"*Thus, the increase in the contribution of O3s in the future is slightly smaller in the RCP8.5 scenario than in the RCP6.0 scenario, despite the larger increase in OMF shown in Figure 2. Here, the different evolution of tropospheric ozone production in the two GHG scenarios plays a crucial role.*"

P22L504 correct to '. . . will consist of 46% ozone from. . . '

Done